# Parametric Complexity Bounds for Approximating PDEs with Neural Networks

**Tanya Marwah, Zachary C. Lipton, Andrej Risteski**
Machine Learning Department, Carnegie Mellon University
{tmarwah, zlipton, aristesk}@andrew.cmu.edu

## Abstract

Recent experiments have shown that deep networks can approximate solutions to high-dimensional PDEs, seemingly escaping the curse of dimensionality. However, questions regarding the theoretical basis for such approximations, including the required network size remain open. In this paper, we investigate the representational power of neural networks for approximating solutions to linear elliptic PDEs with Dirichlet boundary conditions. We prove that when a PDE's coefficients are representable by small neural networks, the parameters required to approximate its solution scale polynomially with the input dimension $d$ and proportionally to the parameter counts of the coefficient networks. To this end, we develop a proof technique that simulates gradient descent (in an appropriate Hilbert space) by growing a neural network architecture whose iterates each participate as sub-networks in their (slightly larger) successors, and converge to the solution of the PDE. We bound the size of the solution showing a polynomial dependence on $d$ and no dependence on the volume of the domain.

## 1 Introduction

A partial differential equation (PDE) relates a multivariate function defined over some domain to its partial derivatives. Typically, one's goal is to solve for the (unknown) function, often subject to additional constraints, such as the function's value on the boundary of the domain. PDEs are ubiquitous in both the natural and social sciences, where they model such diverse processes as heat diffusion [7, 32], fluid dynamics [2, 39], and financial markets [6, 10]. Because most PDEs of interest lack closed-form solutions, computational approximation methods remain a vital and an active field of research [1]. For low-dimensional functions, dominant approaches include the finite differences and finite element methods [26], which discretize the domain. After partitioning the domain into a *mesh*, these methods solve for the function value at the vertices of the mesh. However, these techniques scale exponentially with the input dimension, rendering them unsuitable for high-dimensional problems.

Following breakthroughs in deep learning for approximating high-dimensional functions in such diverse domains as computer vision [21, 33] and natural language processing [3, 9, 40], a burgeoning line of research leverages neural networks to approximate solutions to PDEs. This line of work has produced promising empirical results for common PDEs such as the Hamilton-Jacobi-Bellman and Black-Scholes equations [17, 16, 37]. Because they do not explicitly discretize the domain, and given their empirical success on high-dimensional problems, these methods *appear* not to suffer the curse of dimensionality. However, these methods are not well understood theoretically, leaving open questions about when they are applicable, what their performance depends on, and just how many parameters are required to approximate the solution to a given PDE.

Over the past three years, several theoretical works have investigated questions of representational power under various assumptions. Exploring a variety of settings, [22], [16], and [19], proved that the number of parameters required to approximate a solution to a PDE exhibits a less than exponential

dependence on the input dimension for some special parabolic PDEs that admit straightforward analysis. [15] consider elliptic PDEs with Dirichlet boundary conditions. However, their rate depends on the volume of the domain, and thus can have an implicit exponential dependence on dimension (e.g., consider a hypercube with side length greater than one).

In this paper, we focus on linear elliptic PDEs with Dirichlet boundary conditions, which are prevalent in science and engineering (*e.g.*, the Laplace and Poisson equations). Notably, linear elliptic PDEs define the steady state of processes like heat diffusion and fluid dynamics. Our work asks:

**Question.** *How many parameters suffice to approximate solutions to a linear elliptic PDE up to a specified level of precision using a neural network?*

We show that when the coefficients of the PDE are expressible as small neural networks (note that PDE coefficients are functions), the number of parameters required to approximate the PDE's solution is proportional to the number of parameters required to express the coefficients. Furthermore, we show that the number of parameters depends polynomially on the dimension and does not depend upon the volume of the domain.

## 2 Overview of Results

To begin, we formally define linear elliptic PDEs.

**Definition 1** (Linear Elliptic PDE [11]). *Linear elliptic PDEs with Dirichlet boundary condition can be expressed in the following form:*

$$\begin{cases} (Lu)(x) \equiv (-div(A\nabla u) + cu)(x) = f(x), \forall x \in \Omega, \\ u(x) = 0, \forall x \in \partial\Omega, \end{cases}$$

*where $\Omega \subset \mathbb{R}^d$ is a bounded open set with a boundary $\partial\Omega$. Further, for all $x \in \Omega$, $A : \Omega \to \mathbb{R}^{d \times d}$ is a matrix-valued function, s.t. $A(x) \succ 0$, and $c : \Omega \to \mathbb{R}$, s.t. $c(x) > 0$.* [1]

We refer to $A$ and $c$ as the *coefficients* of the PDE. The divergence form in Definition 1 is one of two canonical ways to define a linear elliptic PDE [11] and is convenient for several technical reasons (see Section 4). The Dirichlet boundary condition states that the solution takes a constant value (here 0) on the boundary $\partial\Omega$.

Our goal is to express the number of parameters required to approximate the solution of a PDE in terms of those required to approximate its coefficients $A$ and $c$. Our key result shows:

**Theorem** (Informal). *If the coefficients $A, c$ and the function $f$ are approximable by neural networks with at most $N$ parameters, the solution $u^\star$ to the PDE in Definition 1 is approximable by a neural network with $O\left(d^{\log(\frac{1}{\epsilon})} N\right)$ parameters.*

This result, formally expressed in Section 5, may help to explain the practical efficacy of neural networks in approximating solutions to high-dimensional PDEs with boundary conditions [37, 27]. To establish this result, we develop a constructive proof technique that simulates gradient descent (in an appropriate Hilbert space) through the very architecture of a neural network. Each iterate, given by a neural network, is subsumed into the (slightly larger) network representing the subsequent iterate. The key to our analysis is to bound both (i) the growth in network size across consecutive iterates; and (ii) the total number of iterates required.

**Organization of the paper** We introduce the required notation along with some mathematical preliminaries on PDEs in Section 4. The problem setting and formal statement of the main result are provided in Section 5. Finally, we provide the proof of the main result in Section 6.

## 3 Prior Work

Among the first papers to leverage neural networks to approximate solutions to PDEs with boundary conditions are [23], [24], and [29]. However, these methods discretize the input space and thus are not

---

[1] Here, div denotes the divergence operator. Given a vector field $F : \mathbb{R}^d \to \mathbb{R}^d$, $div(F) = \nabla \cdot F = \sum_{i=1}^d \frac{\partial F_i}{\partial x_i}$

suitable for high-dimensional input spaces. More recently, mesh-free neural network approaches have been proposed for high-dimensional PDEs [17, 34, 35], achieving impressive empirical results in various applications. [37] design a loss function that penalizes failure to satisfy the PDE, training their network on minibatches sampled uniformly from the input domain. They also provide a universal approximation result, showing that for sufficiently regularized PDEs, there exists a multilayer network that approximates its solution. However, they do not comment on the complexity of the neural network or how it scales with the input dimension. [20] also prove universal approximation power, albeit with networks of size exponential in the input dimension. Recently, [16, 19] provided a better-than-exponential dependence on the input dimension for some special parabolic PDEs, for which the simulating a PDE solver by a neural network is straightforward.

Several recent works [5, 22, 28, 27] show (experimentally) that a single neural network can solve for an entire family of PDEs. They approximate the map from a PDE's parameters to its solution, potentially avoiding the trouble of retraining for every set of coefficients. Among these, only [22] provides theoretical grounding. However, they assume the existence of a finite low-dimensional space with basis functions that can approximate this parametric map—and it is unclear when this would obtain. Our work proves the existence of such maps, under the assumption that the family of PDEs has coefficients described by neural networks with a fixed architecture (Section 7).

In the work most closely related to ours, [15] provides approximation rates polynomial in the input dimension $d$ for the Poisson equation (a special kind of linear elliptic PDE) with Dirichlet boundary conditions. They introduce a walk-on-the-sphere algorithm, which simulates a stochastic differential equation that can be used to solve a Poisson equation with Dirichlet boundary conditions (see, e.g. [31]'s Theorem 9.13). The rates provided in [15] depend on the volume of the domain, and thus depend, implicitly, exponentially on the input dimension $d$. Our result considers the boundary condition for the PDE and is independent of the volume of the domain. Further, we note that our results are defined for a more general linear elliptic PDE, of which the Poisson equation is a special case.

## 4   Notation and Definitions

We now introduce several key concepts from PDEs and some notation. For any open set $\Omega \subset \mathbb{R}^d$, we denote its boundary by $\partial\Omega$ and denote its closure by $\bar{\Omega} := \Omega \cup \partial\Omega$. By $C^0(\Omega)$, we denote the space of real-valued continuous functions defined over the domain $\Omega$. Furthermore, for $k \in \mathbb{N}$, a function $g$ belongs to $C^k(\Omega)$ if all partial derivatives $\partial^\alpha g$ exist and are continuous for any multi-index $\alpha$, such that $|\alpha| \leq k$. Finally, a function $g \in C^\infty(\Omega)$ if $g \in C^k(\Omega)$ for all $k \in \mathbb{N}$. Next, we define several relevant function spaces:

**Definition 2.** *For any $k \in \mathbb{N} \cup \{\infty\}$, $C_0^k(\Omega) := \{g : g \in C^k(\Omega), \overline{supp(g)} \subset \Omega\}$.*

**Definition 3.** *For a domain $\Omega$, the function space $L^2(\Omega)$ consists of all functions $g : \Omega \to \mathbb{R}$, s.t. $\|g\|_{L^2(\Omega)} < \infty$ where $\|g\|_{L^2(\Omega)} = \left(\int_\Omega |g(x)|^2 dx\right)^{\frac{1}{2}}$. This function space is equipped with the inner product*

$$\langle g, h \rangle_{L^2(\Omega)} = \int_\Omega g(x)h(x)dx.$$

**Definition 4.** *For a domain $\Omega$ and a function $g : \Omega \to \mathbb{R}$, the function space $L^\infty(\Omega)$ is defined analogously, where $\|g\|_{L^\infty(\Omega)} = \inf\{c \geq 0 : |g(x)| \leq c \text{ for almost all } x \in \Omega\}$.*

**Definition 5.** *For a domain $\Omega$ and $m \in \mathbb{N}$, we define the Hilbert space $H^m(\Omega)$ as*

$$H^m(\Omega) := \{g : \Omega \to \mathbb{R} : \partial^\alpha g \in L^2(\Omega), \ \forall \alpha \text{ s.t. } |\alpha| \leq m\}$$

*Furthermore, $H^m(\Omega)$ is equipped with the inner product, $\langle g, h \rangle_{H^m(\Omega)} = \sum_{|\alpha| \leq m} \int_\Omega (\partial^\alpha g)(\partial^\alpha h)dx$ and the corresponding norm*

$$\|g\|_{H^m(\Omega)} = \left(\sum_{|\alpha| \leq m} \|\partial^\alpha g\|_{L^2(\Omega)}^2\right)^{\frac{1}{2}}.$$

**Definition 6.** *The closure of $C_0^\infty(\Omega)$ in $H^m(\Omega)$ is denoted by $H_0^m(\Omega)$.*

Informally, $H_0^m(\Omega)$ is the set of functions belonging to $H^m(\Omega)$ that can be approximated by a sequence of functions $\phi_n \in C_0^\infty(\Omega)$. This also implies that if a function $g \in H_0^m(\Omega)$, then $g(x) = 0$ for all $x \in \partial\Omega$. This space (particularly with $m = 1$) is often useful when analyzing elliptic PDEs with Dirichlet boundary conditions.

**Definition 7** (Weak Solution). *Given the PDE in Definition 1, if $f \in L^2(\Omega)$, then a function $u : \Omega \to \mathbb{R}$ solves the PDE in a weak sense if $u \in H_0^1(\Omega)$ and for all $v \in H_0^1(\Omega)$, we have*

$$\int_\Omega (A\nabla u \cdot \nabla v + cuv)\, dx = \int_\Omega fv dx \tag{1}$$

The left hand side of (1) is also equal to $\langle Lu, v \rangle_{L^2(\Omega)}$ for all $u, v \in H_0^1(\Omega)$ (see Lemma 6), whereas, following the definition of the $L^2(\Omega)$ norm, the right side is simply $\langle f, v \rangle_{L^2(\Omega)}$. Having introduced these preliminaries, we now introduce some important facts about linear PDEs that feature prominently in our analysis.

**Proposition 1.** *For the PDE in Definition 1, if $f \in L^2(\Omega)$ the following hold:*

1. *The solution to Equation (1) exists and is unique.*

2. *The weak solution is also the unique solution of the following minimization problem:*

$$u^\star = \underset{v \in H_0^1(\Omega)}{\operatorname{argmin}} J(v) := \underset{v \in H_0^1(\Omega)}{\operatorname{argmin}} \left\{ \frac{1}{2}\langle Lv, v \rangle_{L^2(\Omega)} - \langle f, v \rangle_{L^2(\Omega)} \right\}. \tag{2}$$

This proposition is standard (we include a proof in the Appendix, Section A.1 for completeness) and states that there exists a unique solution to the PDE (referred to as $u^\star$), which is also the solution we get from the variational formulation in (2). In this work, we introduce a sequence of functions that minimizes the loss in the variational formulation.

**Definition 8** (Eigenvalues and Eigenfunctions, [11]). *Given an operator $L$, the tuples $(\lambda, \varphi)_{i=1}^\infty$, where $\lambda_i \in \mathbb{R}$ and $\varphi_i \in H_0^1(\Omega)$ are (eigenvalue, eigenfunction) pairs that satisfy $L\varphi = \lambda\varphi$, for all $x \in \Omega$. Since $\varphi \in H_0^1(\Omega)$, we know that $\varphi_{|\partial\Omega} = 0$. The eigenvalue can be written as*

$$\lambda_i = \inf_{u \in X_i} \frac{\langle Lu, u \rangle_{L^2(\Omega)}}{\|u\|_{L^2(\Omega)}^2}, \tag{3}$$

*where $X_i := span\{\varphi_1, \ldots, \varphi_i\}^\perp = \{u \in H_0^1(\Omega) : \langle u, \varphi_j \rangle_{L^2(\Omega)} = 0 \ \forall j \in \{1, \cdots, i\}\}$ and $0 < \lambda_1 \le \lambda_2 \le \cdots$. Furthermore, we define by $\Phi_k$ the span of the first $k$ eigenfunctions of $L$, i.e., $\Phi_k := span\{\varphi_1, \cdots, \varphi_k\}$.*

We note that since the operator $L$ is self-adjoint and elliptic (in particular, $L^{-1}$ is compact), the eigenvalues are real and countable. Moreover, the eigenfunctions form an orthonormal basis of $H_0^1(\Omega)$ (see [11], Section 6.5).

# 5 Main Result

Before stating our results, we provide the formal assumptions on the PDEs of interest:

**Assumptions:**

(i) **Smoothness**: We assume that the coefficient $A \in \Omega \to \mathbb{R}^{d \times d}$ is a symmetric matrix valued function, i.e., $A = (a_{ij}(x))$ and $a_{ij}(x) \in L^\infty(\Omega)$ is three times differentiable for all $i, j \in [d]$. Furthermore, the function $c \in L^\infty(\Omega)$ is twice differentiable and $c(x) \ge \zeta > 0$ for all $x \in \Omega$. We define a constant $C := (2d^2 + 1) \max\left\{\max_{\alpha:|\alpha|\le 3} \max_{i,j} \|\partial^\alpha a_{ij}\|_{L^\infty(\Omega)}, \max_{\alpha:|\alpha|\le 2} \|\partial^\alpha c\|_{L^\infty(\Omega)}\right\}$. Further, the function $f \in L^2(\Omega)$ is infinitely differentiable and we assume there exists a function $f_{span} \in \Phi_k$, such that for any multi-index $\alpha$, $\|\partial^\alpha f - \partial^\alpha f_{span}\|_{L^2(\Omega)} \le \epsilon_{span}$.

(ii) **Ellipticity**: There exist constants $M \ge m > 0$ such that, for all $x \in \Omega$ and $\xi \in \mathbb{R}^d$,

$$m\|\xi\|^2 \le \sum_{i,j=1}^d a_{ij}(x)\xi_i\xi_j \le M\|\xi\|^2.$$

(iii) **Neural network approximability**: There exist neural networks $\tilde{A}$ and $\tilde{c}$ with $N_A, N_c \in \mathbb{N}$ parameters, respectively, that approximate the functions $A$ and $c$, i.e., $\|A - \tilde{A}\|_{L^\infty(\Omega)} \leq \epsilon_A$ and $\|c - \tilde{c}\|_{L^\infty(\Omega)} \leq \epsilon_c$, for small $\epsilon_A, \epsilon_c \geq 0$. We assume that for all $u \in H_0^1(\Omega)$ the operator $\tilde{L}$ defined as,

$$\tilde{L}u = -\text{div}(\tilde{A}\nabla u) + \tilde{c}u. \tag{4}$$

is elliptic with $(\tilde{\lambda}_i, \tilde{\varphi}_i)_{i=1}^\infty$ (eigenvalue, eigenfunction) pairs. We also assume that there exists a neural network $f_{\text{nn}} \in C^\infty$ with $N_f \in \mathbb{N}$ parameters such that for any multi-index $\alpha$, $\|\partial^\alpha f - \partial^\alpha f_{\text{nn}}\|_{L^2(\Omega)} \leq \epsilon_{\text{nn}}$. By $\Sigma$, we denote the set of all (infinitely differentiable) activation functions used by networks $\tilde{A}$, $\tilde{c}$, and $f_{\text{nn}}$. By $\Sigma'$, we denote the set that contains all the $n$-th order derivatives of the activation functions in $\Sigma$, $\forall n \in \mathbb{N}_0$

Intuitively, ellipticity of $L$ in a linear PDE $Lu = f$ is analogous to positive definiteness of a matrix $Q \in \mathbb{R}^d$ in a linear equation $Qx = k$, where $x, k \in \mathbb{R}^d$.

In (iii), we assume that the coefficients $A$ and $c$, and the function $f$ can be approximated by neural networks. While this is true for any smooth functions given sufficiently large $N_A, N_c, N_f$, our results are most interesting when these quantities are small (e.g. subexponential in the input dimension $d$). For many PDEs used in practice, approximating the coefficients using small neural networks is straightforward. For example, in heat diffusion (whose equilibrium is defined by a linear elliptic PDE) $A(x)$ defines the conductivity of the material at point $x$. If the conductivity is constant, then the coefficients can be written as neural networks with $O(1)$ parameters.

Intuitively, our assumption (iii) that there exists an $f_{\text{span}}$ in $H_0^1(\Omega)$ that lies in the span of the low-lying $k$ eigenfunctions of the operator $L$ can be thought of as a smoothness condition on $f$. For instance, if $L = -\Delta$ (the Laplacian operator), the Dirichlet form satisfies $\frac{\langle Lu, u\rangle_{L^2(\Omega)}}{\|u\|_{L^2(\Omega)}^2} = \frac{\|\nabla u\|_{L^2(\Omega)}}{\|u\|_{L^2(\Omega)}}$, so eigenfunctions corresponding to higher eigenvalues tend to exhibit a higher degree of spikiness. The reader can also think of the eigenfunctions corresponding to larger $k$ as Fourier basis functions corresponding to higher frequencies.

Finally, in (i) and (iii), while the requirement that the function pairs $(f, f_{\text{nn}})$ and $(f, f_{\text{span}})$ are close not only in their values, but their derivatives as well is a matter of analytical convenience, our key results do not necessarily depend on this precise assumption. Alternatively, we could replace this assumption with similar (but incomparable) conditions: e.g., we can also assume closeness of the values and a rapid decay of the $L_2$ norms of the derivatives. We require control over the derivatives because our method's gradient descent iterations involve repeatedly applying the operator $L$ to $f$—which results in progressively higher derivatives.

We can now formally state our main result:

**Theorem 1** (Main Theorem). *Consider a linear elliptic PDE satisfying Assumptions (i)-(iii), and let $u^\star \in H_0^1(\Omega)$ denote its unique solution. If there exists a neural network $u_0 \in H_0^1(\Omega)$ with $N_0$ parameters, such that $\|u^\star - u_0\|_{L^2(\Omega)} \leq R$, for some $R < \infty$, then for every $\epsilon > 0$, there exists a neural network $u_\epsilon$ with size*

$$O\left(d^{2T}\left(N_0 + N_A\right) + T(N_f + N_c)\right)$$

*such that $\|u^\star - u_\epsilon\|_{L^2(\Omega)} \leq \epsilon + \tilde{\epsilon}$ where,*

$$\tilde{\epsilon} := \frac{\epsilon_{\text{span}}}{\lambda_1} + \frac{\delta}{\lambda_1}\frac{2^{3/2}\|f\|_{L^2(\Omega)}}{\gamma - \delta} + \delta\|u^\star\|_{L^2(\Omega)} + (\max\{1, T^2 C\eta\})^T\left(\epsilon_{\text{span}} + \epsilon_{nn} + \frac{2^{3/2}\delta\|f\|_{L^2(\Omega)}}{\gamma - \delta}\right)$$

*and $\kappa = \frac{\tilde{\lambda}_k + \tilde{\lambda}_1}{\tilde{\lambda}_k - \tilde{\lambda}_1}$, $\eta = \frac{2}{\tilde{\lambda}_1 + \tilde{\lambda}_k}$, $T = O\left(\frac{\log(R/\epsilon)}{\log \kappa}\right)$, and $\delta = \frac{1}{\min\{m/\epsilon_A, \zeta/\epsilon_c\}}$ and $\alpha$ is a multi-index. Furthermore, the activation functions used in $u_\epsilon$ belong to the set $\Sigma \cup \Sigma' \cup \{\rho\}$ where $\rho(y) = y^2$ for all $y \in \mathbb{R}$ is the square activation function.*

This theorem shows that given an initial neural network $u_0 \in H_0^1(\Omega)$ containing $N_0$ parameters, we can recover a neural network that is $\epsilon$ close to the unique solution $u^\star$. The number of parameters in $u_\epsilon$ depend on how close the initial estimate $u_0$ is to the solution $u^\star$, and $N_0$. This results in a trade-off, where better approximations may require more parameters, compared to a poorer approximation with fewer parameters.

Note that $\epsilon$ can be taken arbitrarily close to $0$, while $\tilde{\epsilon}$ is a "bias" error term that does not go to $0$. The first three terms in the expression for $\tilde{\epsilon}$ result from bounding the difference between the solutions to the equations $Lu = f$ and $\tilde{L}u = f_{\text{span}}$, whereas the third term is due to difference between $f$ and $f_{\text{nn}}$ and the fact that our proof involves simulating the gradient descent updates with neural networks.

The term $T = O\left(\frac{\log(R/\epsilon)}{\log(\kappa)}\right)$ comes from the fact that we are simulating $T$ steps of a gradient descent-like procedure on a strongly convex loss with condition number $\kappa$ to reach an $\epsilon$-approximate optimum. The parameters $\lambda_k$ and $\lambda_1$ can be thought of as the effective Lipschitz and strong-convexity constants of the loss. Finally, to give a sense of what $R$ looks like, we show in Corollary 2 (see Section B in the Appendix) that if $u_0$ is initialized to be identically zero then $R = \frac{\|f\|_{L^2(\Omega)}}{\lambda_1}$.

**We make few remarks about the theorem statement:**

**Remark 1.** *While we state our convergence results in $L^2(\Omega)$ norm, our proof works for the $H_0^1(\Omega)$ norm as well. This is because in the space defined by the top-k eigenfunctions of the operator $L$, $L^2(\Omega)$ and $H_0^1(\Omega)$ norm are equivalent (shown in Proposition 2). Further, note that even though we have assumed that $u^\star \in H_0^1(\Omega)$ is the unique solution of (1) from the boundary regularity condition, we have that $u^\star \in H^2(\Omega)$ (see [11], Theorem 4 in Chapter 6). This ensures that the solution $u^\star$ is twice differentiable as well.*

**Remark 2.** *To get a sense of the scale of $\lambda_1$ and $\lambda_k$, when $L = -\Delta$ (the Laplacian operator), the eigenvalue $\lambda_1 = \inf_{u \in H_0^1(\Omega)} \frac{\|\nabla u\|_{L^2(\Omega)}}{\|u\|_{L^2(\Omega)}} = \frac{1}{C_p}$, where $C_p$ is the Poincaré constant (see Theorem 2 in Appendix). For geometrically well-behaved sets $\Omega$ (e.g. convex sets with a strongly convex boundary, like a sphere), $C_p$ is even dimension-independent. Further from the Weyl's law operator ([11], Section 6.5) we have*

$$\lim_{k \to \infty} \frac{\lambda_k^{d/2}}{k} = \frac{(2\pi)^d}{\text{vol}(\Omega)\alpha(d)}$$

*where $\alpha(d)$ is the volume of a unit ball in $d$ dimensions. So, if $\text{vol}(\Omega) \geq 1/\alpha(d)$, $\lambda_k$ grows as $O(k^{2/d})$, which is a constant so long as $\log k \ll d$.*

**Remark 3.** *The choice of activation functions (in particular, the requirement that $\Sigma$ only contains differentiable functions) is for mathematical convenience. Namely, by standard results from approximation theory [41, 4] one can approximate a neural network with one choice of nonlinearity via a (comparably sized) neural network with another choice of nonlinearity (under mild assumptions) by incurring a dimension-independent increase in size. Thus, if the coefficients are approximable by a neural network with a non-differentiable activation function, they can also be approximated by a slightly larger network with a differentiable activation over any compact domain. Similarly, the activations on the network resulting from Theorem 1 can be replaced by a different activation at the expense of a slight blowup in size. Details are included in Appendix C.4.*

## 6 Proof of Main Result

First, we provide some intuition behind the proof, via an analogy between a uniformly elliptic operator and a positive definite matrix in linear algebra. We can think of finding the solution to the equation $Lu = f$ for an elliptic $L$ as analogous to finding the solution to the linear system of equations $Qx = k$, where $Q$ is a $d \times d$ positive definite matrix, and $x$ and $k$ are $d$-dimensional vectors. One way to solve such a linear system is by minimizing the strongly convex function $\|Qx - b\|^2$ using gradient descent. Since the objective is strongly convex, after $O(\log(1/\epsilon))$ gradient steps, we reach an $\epsilon$-optimal point in an $l_2$ sense.

Our proof uses a similar strategy. First, we show that for the operator $L$, we can define a sequence of functions that converge to an $\epsilon$-optimal function approximation (in this case in the $L^2(\Omega)$ norm) after $O(\log(1/\epsilon))$ steps—similar to the rate of convergence for strongly convex functions. Next, we inductively show that each iterate in the sequence can be approximated by a small neural network. More precisely, we show that given a bound on the size of the $t$-th iterate $u_t$, we can, in turn, upper bound the size of the $(t + 1)$-th iterate $u_{t+1}$ because the update transforming $u_t$ to $u_{t+1}$ can be simulated by a small neural network (Lemma 4). These iterations look roughly like $u_{t+1} \leftarrow u_t - \eta(Lu_t - f)$, and we use a "backpropagation" lemma (Lemma 7) which bounds the size of the derivative of a neural network.

## 6.1 Defining a Convergent Sequence

The rough idea is to perform gradient descent in $L^2(\Omega)$ [30, 12, 13], and define a convergent sequence whose iterates converge to $u^\star$ in $L^2(\Omega)$ norm (and following Remark 1, in $H_0^1(\Omega)$ as well). However, there are two obstacles to defining the iterates as simply $u_{t+1} \leftarrow u_t - \eta(Lu_t - f)$, (1) $L$ is unbounded—so the standard way of choosing a step size for gradient descent (roughly the ratio of the minimum and maximum eigenvalues of $L$) would imply choosing a step size $\eta = 0$, and (2) $L$ does not necessarily preserve the boundary conditions, so if we start with $u_t \in H_0^1(\Omega)$, it may be that $Lu_t - f$ does not even lie in $H_0^1(\Omega)$.

We resolve both issues by restricting the updates to the span of the first $k$ eigenfunctions of $L$. More concretely, as shown in Lemma 1, if a function $u$ in $\Phi_k$, then the function $Lu$ will also lie in $\Phi_k$. We also show that within the span of the first $k$ eigenfunctions, $L$ is bounded (with maximum eigenvalue $\lambda_k$), and can therefore be viewed as an operator from $\Phi_k$ to $\Phi_k$. Further, we use $f_{\text{span}}$ instead of $f$ in our updates, which now have the form $u_{t+1} \leftarrow u_t - \eta(Lu_t - f_{\text{span}})$. Since $f_{\text{span}}$ belongs to $\Phi_k$, for a $u_t$ in $\Phi_k$ the next iterate $u_{t+1}$ will now remain in $\Phi_k$. Continuing the matrix analogy, we can choose the usual step size of $\eta = \frac{2}{\lambda_1 + \lambda_k}$. Precisely, we show:

**Lemma 1.** *Let $L$ be an elliptic operator. Then, for all $v \in \Phi_k$ it holds:*

1. *$Lv \in \Phi_k$.*

2. *$\lambda_1 \|v\|_{L^2(\Omega)} \leq \langle Lv, v \rangle_{L^2(\Omega)} \leq \lambda_k \|v\|_{L^2(\Omega)}$*

3. *$\left\| \left(I - \frac{2}{\lambda_k + \lambda_1} L\right) u \right\|_{L^2(\Omega)} \leq \frac{\lambda_k - \lambda_1}{\lambda_k + \lambda_1} \|u\|_{L^2(\Omega)}$*

The proof of this lemma is provided in Section C.1.

Further, note that we will use a slight variant of the updates and instead set $u_{t+1} \leftarrow u_t - \eta(\tilde{L}u - \tilde{f}_{\text{span}})$ as the iterates of the convergent sequence. Here, the operator $\tilde{L}$ (defined in (4)) has the neural network approximations of $A$ and $c$ as its coefficients, and $\tilde{f}_{\text{span}}$ is a function that lies in span of the first $k$ eigenfunctions of $\tilde{L}$ (denoted by $\tilde{\Phi}_k$), such that $\|f_{\text{span}} - \tilde{f}_{\text{span}}\|_{L^2(\Omega)}$ is small (for an exact statement, see Lemma 11 in the Appendix). In Section 6.2, we will see that updates defined thusly will be more convenient to simulate via a neural network.

The sequence defined so far satisfies two important properties. First, the convergence point of the sequence and $u^*$, the solution to the original PDE, are not too far from each other. Concretely:

**Lemma 2.** *Assume that $\tilde{u}^\star_{\text{span}}$ is the solution to the PDE $\tilde{L}u = \tilde{f}_{\text{span}}$, where $\tilde{f}_{\text{span}} : H_0^1(\Omega) \to \mathbb{R}$ and $\tilde{f}_{\text{span}} \in \tilde{\Phi}_k$. Given Assumptions (i)-(iii), we have $\|u^\star - \tilde{u}^\star_{\text{span}}\|_{L^2(\Omega)} \leq \epsilon$, such that $\epsilon = \frac{\epsilon_{\text{span}}}{\lambda_1} + \frac{\delta}{\lambda_1} \frac{2^{3/2} \|f\|_{L^2(\Omega)}}{\gamma - \delta} + \delta \|\tilde{u}^\star_{\text{span}}\|_{L^2(\Omega)}$, where $\gamma = \frac{1}{\lambda_k} - \frac{1}{\lambda_{k+1}}$ and $\delta = \frac{1}{\min\{m/\epsilon_A, \zeta/\epsilon_c\}}$.*

The proof for Lemma 2 is provided in the Appendix (Section D.1). Each of the three terms in the final error captures different sources of perturbation: the first term comes from approximating $f$ by $f_{\text{span}}$; the second term comes from applying Davis-Kahan [8] to bound the "misalignment" between the eigenspaces $\Phi_k$ and $\tilde{\Phi}_k$ (hence, the appearance of the eigengap between the $k$ and $(k + 1)$-st eigenvalue of $L^{-1}$); the third term is a type of "relative" error bounding the difference between the solutions to the PDEs $Lu = \tilde{f}_{\text{span}}$ and $\tilde{L}u = \tilde{f}_{\text{span}}$.

The second property of the sequence of functions is that they converge exponentially fast. The rate is characterized in the following lemma:

**Lemma 3** (Convergence of gradient descent in $L^2$)**.** *Let $\tilde{u}^\star_{\text{span}}$ denote the unique solution to the PDE $\tilde{L}u = \tilde{f}_{\text{span}}$, where $\tilde{f}_{\text{span}} \in \tilde{\Phi}_k$, and the operator $\tilde{L}$ satisfies the conditions in Lemma 1. Then for any $u_0 \in H_0^1(\Omega)$ such that $u_0 \in \tilde{\Phi}_k$, we define the sequence*

$$u_{t+1} \leftarrow u_t - \frac{2}{\tilde{\lambda}_1 + \tilde{\lambda}_k}(\tilde{L}u_t - \tilde{f}_{\text{span}}) \quad (t \in \mathbb{N}) \tag{5}$$

where for all $t \in \mathbb{N}$, $u_t \in H_0^1(\Omega)$. *Then for any $\epsilon \geq 0$ we have $\|u_T - \tilde{u}_{\mathrm{span}}^\star\|_{L^2(\Omega)} \leq \epsilon$ after $T$ iterations where,*

$$T \geq \frac{\log\left(\frac{\|u_0 - \tilde{u}_{\mathrm{span}}^\star\|_{L^2(\Omega)}}{\epsilon}\right)}{\log\left(\frac{\tilde{\lambda}_k + \tilde{\lambda}_1}{\tilde{\lambda}_k - \tilde{\lambda}_1}\right)}$$

The proof for Lemma 3 is similar to the analysis of the convergence time of gradient descent for strongly convex losses and can be found in Section C.2 of the Appendix. Finally, combining the results from Lemma 2 and Lemma 3 via triangle inequality, we have:

$$\|u^\star - u_T\|_{L^2(\Omega)} \leq \|u^\star - \tilde{u}_{\mathrm{span}}^\star\|_{L^2(\Omega)} + \|\tilde{u}_{\mathrm{span}}^\star - u_T\|_{L^2(\Omega)}$$

and the first term on the RHS subsumes the first three summands of $\tilde{\epsilon}$ defined in Theorem 1.

## 6.2 Approximating iterates by neural networks

In Lemma 3, we show that there exists a sequence of functions (5) which converge fast to a function close to $u^\star$. The next step in the proof is to approximate the iterates by neural networks.

The main idea is as follows. Suppose first the iterates $u_{t+1} = u_t - \eta(\tilde{L}u_t - \tilde{f}_{\mathrm{span}})$ are such that $\tilde{f}_{\mathrm{span}}$ is exactly representable as a neural network. Then, the iterate $u_{t+1}$ can be written in terms of three operations performed on $u_t$, $a$ and $f$: taking derivatives, multiplication and addition. Moreover, if $g$ is representable as a neural network with $N$ parameters, the coordinates of the vector $\nabla g$ can be represented by a neural network with $O(N)$ parameters. This is a classic result (Lemma 7), essentially following from the backpropagation algorithm. Finally, addition or multiplication of two functions representable as neural networks with sizes $N_1, N_2$ can be represented as neural networks with size $O(N_1 + N_2)$ (see Lemma 8).

Using these facts, we can write down a recurrence upper bounding the size of neural network approximation $u_{t+1}$, denoted by $\hat{u}_{t+1}$, in terms of the number of parameters in $\hat{u}_t$ (which is the neural network approximation to $u_t$). Formally, we have:

**Lemma 4** (Recursion Lemma). *Given the Assumptions (i)-(iii), consider the update equation*

$$\hat{u}_{t+1} \leftarrow \hat{u}_t - \frac{2}{\tilde{\lambda}_1 + \tilde{\lambda}_k}\left(\tilde{L}\hat{u}_t - f_{\mathrm{nn}}\right) \tag{6}$$

*If at step $t$, $\hat{u}_t : \mathbb{R}^d \to \mathbb{R}$ is a neural network with $N_t$ parameters, then the function $\hat{u}_{t+1}$ is a neural network with $O(d^2(N_A + N_t) + N_t + N_{\tilde{f}} + N_c)$ parameters.*

*Proof.* Expand the update $\hat{u}_{t+1} \leftarrow \hat{u}_t - \eta\left(\tilde{L}\hat{u}_t - f_{\mathrm{nn}}\right)$ as follows:

$$\hat{u}_{t+1} \leftarrow \hat{u}_t - \eta\left(\sum_{i,j=1}^d \tilde{a}_{ij}\partial_{ij}\hat{u}_t + \sum_{j=1}^d \left(\sum_{i=1}^d \partial_i \tilde{a}_{ij}\right)\partial_j \hat{u}_t + \tilde{c}\hat{u}_t - f_{\mathrm{nn}}\right).$$

Using Lemma 7, $\partial_{ij}\hat{u}_t$, $\partial_j \hat{u}_t$ and $\partial_i \tilde{a}_{ij}$ can be represented by a neural network with $O(N_t)$, $O(N_t)$ and $O(N_A)$ parameters, respectively. Further, $\partial_i \tilde{a}_{ij}\partial_j u$ and $\tilde{a}_{ij}\partial_{ij}\hat{u}$ can be represented by a neural network with $O(N_A + N_t)$ parameters, and $\tilde{c}\hat{u}_t$ can be represented by a network with $O(N_t + N_c)$ parameters, from Lemma 8. Hence $\hat{u}_{t+1}$ can be represented in $O(d^2(N_A + N_t) + N_f + N_c + N_t)$ parameters. Note that, throughout the entire proofs $O$ hides independent constants. $\square$

Combining the results of Lemma 3 and Lemma 4, we can get a recurrence for the number of parameters required to represent the neural network $\hat{u}_t$:

$$N_{t+1} \leq d^2 N_t + d^2 N_A + N_t + N_{\tilde{f}} + N_c$$

Unfolding this recurrence, we get $N_T \leq d^{2T}N_0 + \frac{d^2(d^T-1)}{d^2-1}N_A + T(N_f) + N_c)$.

Hence, the total number of parameters required for a neural network to approximate a solution to a PDE of the form in Definition 1

$$O\left(d^{2\frac{\log\left(\frac{\|u_0 - \tilde{u}_{\mathrm{span}}^\star\|_{L^2(\Omega)}}{\epsilon}\right)}{\log \kappa}}(N_0 + N_A) + \frac{\log\left(\frac{\|u_0 - \tilde{u}_{\mathrm{span}}^\star\|_{L^2(\Omega)}}{\epsilon}\right)}{\log \kappa}(N_f + N_c)\right)$$

Finally, we have to deal with the fact that $\tilde{f}_{\text{span}}$ is not exactly a neural network, but only approximately so. The error due to this discrepancy can be characterized through the following lemma:

**Lemma 5** (Error using $f_{\text{nn}}$)**.** *Consider the update equation in* (6)*, where $f_{\text{nn}}$ is a neural network with $N_f$. Then the neural network $\hat{u}_t$ approximates the function $u_t$ such that $\|u_t - \hat{u}_t\|_{L^2(\Omega)} \le \epsilon_{\text{nn}}^{(t)}$ where $\epsilon_{\text{nn}}^{(t)}$ is*

$$O\left( (\max\{1, t^2 \eta eC\})^t \left( \epsilon_{\text{span}} + \epsilon_{\text{nn}} + \frac{2^{3/2} \delta \|f\|_{L^2(\Omega)}}{\gamma - \delta} \right) \right)$$

*where $\delta = \frac{1}{\min\{m/\epsilon_A, \zeta/\epsilon_c\}}$, $\gamma = \frac{1}{\lambda_k} - \frac{1}{\lambda_{k+1}}$, and $\alpha$ is a multi-index.*

The main strategy to prove this lemma involves tracking the "residual" non-neural-network part of the iterates. Precisely, we can write the update $u_{t+1} = u_t - \eta(\tilde{L}u_t - (f_{\text{nn}} + r))$, for a "residual" function $r = \tilde{f}_{\text{span}} - f_{\text{nn}}$. If $u_t$ was exactly a neural network, the first part of the update, $u_t - \eta(\tilde{L}u_t - f_{\text{nn}})$ can be written as a neural network as in Lemma 4, and $\eta r$ can be treated as an error. Thus, in order to bound the error for using $f_{\text{nn}}$ instead of $\tilde{f}_{\text{span}}$, we maintain a decomposition of $u_t$ into a neural network part, and a residual part, and inductively bound the total residual part. Given the recurrent structure of our updates, at each iteration $t$, there will be an accumulation of the earlier $t-1$ applications of the operator $\tilde{L}$, that results in an increasing number of higher order derivatives of $r$ to be bounded at each step. This is why we require that $f_{\text{nn}}$ is close to $f$ not only in the $L_2$ sense but also in terms of their higher order derivatives.

## 7 Applications to Learning Operators

A number of recent works attempt to simultaneously approximate the solutions for an entire family of PDEs by learning a *parametric map* that takes as inputs (some representation of) the coefficients of a PDE and returns its solution [5, 28, 27]. For example, given a set of observations that $\{a_j, u_j\}_{j=1}^N$, where each $a_j$ denotes a coefficient of a PDE with corresponding solution $u_j$, they learn a neural network $G$ such that for all $j$, $u_j = G(a_j)$. Our parametric results provide useful insights for why simultaneously solving an entire family of PDEs with a *single neural network $G$* is possible in the case of linear elliptic PDEs.

Consider the case where the coefficients $a_j$ in the family of PDEs are given by neural networks with a fixed architecture, but where each instance of a PDE is characterized by a different setting of the weights in the models representing the coefficients. Lemma 4 shows that each iteration of our sequence (5) constructs a new network containing both the current solution and the coefficient networks as subnetworks. We can view our approximation as not merely approximating the solution to a single PDE but to every PDE in the family, by treating the coefficient networks as placeholder architectures whose weights are provided as inputs. Thus, our construction provides a parametric map between the coefficients of an elliptic PDE in this family and its solution.

## 8 Conclusion and Future Work

We derive parametric complexity bounds for neural network approximations for solving linear elliptic PDEs with Dirichlet boundary conditions, whenever the coefficients can be approximated by are neural networks with finite parameter counts. By simulating gradient descent in function spaces using neural networks, we construct a neural network that approximates the solution of a PDE. We show that the number of parameters in the neural network depends on the parameters required to represent the coeffcients and has a poly$(d)$ dependence on the dimension of the input space, therefore avoiding the curse of dimensionality.

An immediate open question is related to the tightening our results: our current error bound is sensitive to the neural network approximation lying close to $\Phi_k$ which could be alleviated by relaxing (by adding some kind of "regularity" assumptions) the dependence of our analysis on the first $k$ eigenfunctions. Further, the dependencies in the exponent of $d$ on $R$ and $\kappa$ in parametric bound may also be improvable. Finally, the idea of simulating an iterative algorithm by a neural network to derive a representation-theoretic result is broadly applicable, and may be a fertile ground for further work, both theoretically and empirically, as it suggest a particular kind of weight tying.

# 9 Acknowledgement

This paper is based upon work funded and supported by the Department of Defense under contract FA8702-15-D-0002

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
