# A Brief Overview of Partial Differential Equations

In this section, we introduce few key definitions and results from PDE literature. We note that the results in this section are standard and have been included in the Appendix for completeness. We refer the reader to classical texts on PDEs [11, 14] for more details.

We will use the following Poincaré inequality throughout our proofs.

**Theorem 2** (Poincaré's inequality). *Given $\Omega \subset \mathbb{R}^d$, a bounded open subset, there exists a constant $C_p > 0$ such that for all $u \in H_0^1(\Omega)$*

$$\|u\|_{L^2(\Omega)} \leq C_p \|\nabla u\|_{L^2(\Omega)}.$$

**Corollary 1.** *For the bounded open subset $\Omega \subset \mathbb{R}^d$, for all $u \in H_0^1(\Omega)$, we define the norm in the Hilbert space $H_0^1(\Omega)$ as*

$$\|u\|_{H_0^1(\Omega)} = \|\nabla u\|_{L^2(\Omega)}. \tag{7}$$

*Further, the norm in $H_0^1(\Omega)$ is equivalent to the norm $H^1(\Omega)$.*

*Proof.* Note that for $u \in H_0^1(\Omega)$ we have,

$$\|u\|_{H^1(\Omega)} = \|\nabla u\|_{L^2(\Omega)} + \|u\|_{L^2(\Omega)}$$
$$\geq \|\nabla u\|_{L^2(\Omega)}$$
$$\implies \|u\|_{H^1(\Omega)} \geq \|u\|_{H_0^1(\Omega)}.$$

Where we have used the definition of the norm in $H_0^1(\Omega)$ space.

Further, using the result in Theorem 2 we have

$$\|u\|_{H^1(\Omega)}^2 = \left(\|u\|_{L^2(\Omega)}^2 + \|\nabla u\|_{L^2(\Omega)}^2\right) \leq \left(C_p^2 + 1\right)\|\nabla u\|_{H^1(\Omega)}^2 \tag{8}$$

Therefore, combining the two inequalities we have

$$\|u\|_{H_0^1(\Omega)} \leq \|u\|_{H^1(\Omega)} \leq C_h \|u\|_{H_0^1(\Omega)} \tag{9}$$

where $C_h = (C_p^2 + 1)$. Hence we have that the norm in $H_0^1(\Omega)$ and $H^1(\Omega)$ spaces are equivalent. $\square$

**Proposition 2** (Equivalence between $L^2(\Omega)$ and $H_0^1(\Omega)$ norms). *If $v \in \Phi_k$ then we have that $\|v\|_{L^2(\Omega)}$ is equivalent to $\|v\|_{H_0^1(\Omega)}$.*

*Proof.* We have from the Poincare inequality in Theorem 2 that for all $v \in H_0^1(\Omega)$, the norm in $L^2(\Omega)$ is upper bounded by the norm in $H_0^1(\Omega)$, i.e.,

$$\|v\|_{L^2(\Omega)}^2 \leq \|v\|_{H_0^1(\Omega)}^2$$

Further, using results from (11) and (10) (where $b(u,v) := \langle Lu, v\rangle_{L^2(\Omega)}$), we know that for all $v \in H_0^1(\Omega)$ we have

$$m\|v\|_{H_0^1(\Omega)}^2 \leq \langle Lv, v\rangle_{L^2(\Omega)} \leq \max\{M, C_p\|c\|_{L^\infty(\Omega)}\}\|v\|_{H_0^1(\Omega)}^2$$

This implies that $\langle Lu, v\rangle_{L^2(\Omega)}$ is equivalent to the inner product $\langle u, v\rangle_{H_0^1(\Omega)}$, i.e., for all $u, v \in H_0^1(\Omega)$,

$$m\langle u, v\rangle_{H_0^1(\Omega)} \leq \langle Lu, v\rangle_{L^2(\Omega)} \leq \max\left\{M, C_p\|c\|_{L^\infty(\Omega)}\right\}\langle u, v\rangle_{H_0^1(\Omega)}$$

Further, since $v \in \Phi_k$, we have from Lemma 1 that

$$\langle Lv, v\rangle_{L^2(\Omega)} \leq \lambda_k \|v\|_{L^2(\Omega)}^2$$

$$\implies \|v\|_{H_0^1(\Omega)} \leq \frac{\lambda_k}{c_1}\|v\|_{L^2(\Omega)}^2$$

Hence we have that for all $v \in \Phi_k$ $\|v\|_{L^2(\Omega)}$ is equivalent to $\|v\|_{H_0^1(\Omega)}$ and by Corollary 1 is also equivalent to $\|v\|_{H^1(\Omega)}$. $\square$

Now introduce a form for $\langle Lu, v \rangle_{L^2(\Omega)}$ that is more amenable for the existence and uniqueness results.

**Lemma 6.** *For all $u, v \in H_0^1(\Omega)$, we have the following,*

1. *The inner product $\langle Lu, v \rangle_{L^2(\Omega)}$ equals,*

$$\langle Lu, v \rangle_{L^2(\Omega)} = \int_\Omega \left( A\nabla u \cdot \nabla v + cuv \right) \, dx$$

2. *The operator L is self-adjoint.*

*Proof.* 1. We will be using the following integration by parts formula,

$$\int_\Omega \frac{\partial u}{\partial x_i} dx = - \int_\Omega u \frac{\partial v}{\partial x_i} dx + \int_{\partial\Omega} uvn_i \partial\Gamma$$

Where $n_i$ is a normal at the boundary and $\partial\Gamma$ is an infinitesimal element of the boundary.

Hence we have for all $u, v \in H_0^1(\Omega)$,

$$\langle Lu, v \rangle_{L^2(\Omega)} = \int_\Omega - \left( \sum_{i=1}^{d} \left( \partial_i \left( A\nabla u \right)_i \right) \right) v + cuv \, dx$$

$$= \int_\Omega A\nabla u \cdot \nabla v dx - \int_{\partial\Omega} \left( \sum_{i=1}^{d} (A\nabla u)_i n_i \right) vd\Gamma + \int_\Omega cuv dx$$

$$= \int_\Omega A\nabla u \cdot \nabla v dx + \int_\Omega cuv dx \qquad (\because v_{|\partial\Omega} = 0)$$

2. To show that the operator $L : H_0^1(\Omega) \to H_0^1(\Omega)$ is self-adjoint, we show that for all $u, v \in H_0^1(\Omega)$ we have $\langle Lu, v \rangle = \langle u, Lv \rangle$.

From Proposition 6, for functions $u, v \in H_0^1(\Omega)$ we have

$$\langle Lu, v \rangle_{L^2(\Omega)} = \int_\Omega A\nabla u \cdot \nabla v dx + \int_\Omega cuv dx$$

$$= \int_\Omega A\nabla v \cdot \nabla u dx + \int_\Omega cvu dx$$

$$= \langle u, Lv \rangle$$

$\square$

## A.1 Proof of Proposition 1

We first show that if $u$ is the unique solution then it minimizes the variational norm.

Let $u$ denote the weak solution, further for all $w \in H_0^1(\Omega)$ let $v = u + w$. Using the fact that $L$ is self-adjoint (as shown in Lemma 6) we have

$$J(v) = J(u + w) = \frac{1}{2} \langle L(u + w), (u + w) \rangle_{L^2(\Omega)} - \langle f, u + w \rangle_{L^2(\Omega)}$$

$$= \frac{1}{2} \langle Lu, u \rangle_{L^2(\Omega)} + \frac{1}{2} \langle Lw, w \rangle_{L^2(\Omega)} + \langle Lu, w \rangle_{L^2(\Omega)} - \langle f, u \rangle_{L^2(\Omega)} - \langle f, w \rangle_{L^2(\Omega)}$$

$$= J(u) + \frac{1}{2} \langle Lw, w \rangle_{L^2(\Omega)} + \langle Lu, w \rangle_{L^2(\Omega)} - \langle f, w \rangle_{L^2(\Omega)}$$

$$\geq J(u)$$

where we use the fact that $\langle Lu, u \rangle_{L^2(\Omega)} > 0$ and that $u$ is a weak solution hence (1) holds for all $w \in H_0^1(\Omega)$.

To show the other side, assume that $u$ minimizes $J$, i.e., for all $\lambda > 0$ and $v \in H_0^1(\Omega)$ we have, $J(u + \lambda v) \geq J(u)$,

$$J(u + \lambda v) \geq J(u)$$

$$\frac{1}{2}\langle L(u + \lambda v), (u + \lambda v)\rangle_{L^2(\Omega)} - \langle f, (u + \lambda v)\rangle_{L^2(\Omega)} \geq \frac{1}{2}\langle Lu, u\rangle_{L^2(\Omega)} - \langle f, u\rangle_{L^2(\Omega)}$$

$$\implies \frac{\lambda}{2}\langle Lv, v\rangle_{L^2(\Omega)} + \langle Lu, v\rangle_{L^2(\Omega)} - \langle f, v\rangle_{L^2(\Omega)} \geq 0$$

Taking $\lambda \to 0$, we get

$$\langle Lu, v\rangle_{L^2(\Omega)} - \langle f, v\rangle_{L^2(\Omega)} \geq 0$$

and also taking $v$ as $-v$, we have

$$\langle Lu, v\rangle_{L^2(\Omega)} - \langle f, v\rangle_{L^2(\Omega)} \leq 0$$

Together, this implies that if $u$ is the solution to (2), then $u$ is also the weak solution, i.e, for all $v \in H_0^1(\Omega)$ we have

$$\langle Lu, v\rangle_{L^2(\Omega)} = \langle f, v\rangle_{L^2(\Omega)}$$

### Proof for Existence and Uniqueness of the Solution

In order to prove for the uniqueness of the solution, we first state the Lax-Milgram theorem.

**Theorem 3** (Lax-Milgram, [25]). *Let $\mathcal{H}$ be a Hilbert space with inner-product $(\cdot, \cdot) : \mathcal{H} \times \mathcal{H} \to \mathbb{R}$, and let $b : \mathcal{H} \times \mathcal{H} \to \mathbb{R}$ and $l : \mathcal{H} \to \mathbb{R}$ be the bilinear form and linear form, respectively. Assume that there exists constants $C_1, C_2, C_3 > 0$ such that for all $u, v \in \mathcal{H}$ we have,*

$$C_1\|u\|_{\mathcal{H}}^2 \leq b(u, u), \quad |b(u, v)| \leq C_2\|u\|_{\mathcal{H}}\|v\|_{\mathcal{H}}, \quad \text{and } |l(u)| \leq C_3\|u\|_{\mathcal{H}}.$$

*Then there exists a unique $u \in \mathcal{H}$ such that,*

$$b(u, v) = l(v) \quad \text{for all } v \in \mathcal{H}.$$

Having stated the Lax-Milgram Theorem, we make the following proposition,

**Proposition 3.** *Given the assumptions (i)-(iii), solution to the variational formulation in Equation 1 exists and is unique.*

*Proof.* Using the *variational formulation* defined in (1), we introduce the bilinear form $b(\cdot, \cdot) : H_0^1(\Omega) \times H_0^1(\Omega) \to \mathbb{R}$ where $b(u, v) := \langle Lu, v\rangle$. Hence, we prove the theorem by showing that the bilinear form $b(u, v)$ satisfies the conditions in Theorem 3.

We first show that for all $u, v \in H_0^1(\Omega)$ the following holds,

$$
\begin{aligned}
|b(u, v)| &= \left| \int_\Omega (A\nabla u \cdot \nabla v + cuv)\, dx \right| \\
&\leq \int_\Omega |(A\nabla u \cdot \nabla v + cuv)|\, dx \\
&\leq \int_\Omega |A\nabla u \cdot \nabla v|\, dx + \int_\Omega |cuv|\, dx \\
&\leq \|A\|_{L^\infty(\Omega)}\|\nabla u\|_{L^2(\Omega)}\|\nabla v\|_{L^2(\Omega)} + \|c\|_{L^\infty(\Omega)}\|u\|_{L^2(\Omega)}\|v\|_{L^2(\Omega)} \\
&\leq M\|\nabla u\|_{L^2(\Omega)}\|\nabla v\|_{L^2(\Omega)} + \|c\|_{L^\infty(\Omega)}\|u\|_{L^2(\Omega)}\|v\|_{L^2(\Omega)} \\
&\leq \max\left\{M, C_p\|c\|_{L^\infty(\Omega)}\right\} \|u\|_{H_0^1(\Omega)}\|v\|_{H_0^1(\Omega)}
\end{aligned}
\tag{10}
$$

Now we show that the bilinear form $a(u, u)$ is lower bounded.

$$
\begin{aligned}
b(v, v) &= \int_\Omega \left(A\nabla v \cdot \nabla v + cv^2\right) dx \\
&\geq m \int_\Omega \|\nabla v\|^2 dx = m\|v\|_{H_0^1(\Omega)}
\end{aligned}
\tag{11}
$$

Finally, for $v \in H_0^1(\Omega)$

$$|(f, v)| = \left| \int_\Omega f v dx \right| \le \|f\|_{L^2(\Omega)} \|v\|_{L^2(\Omega)} \le C_p \|f\|_{L^2(\Omega)} \|v\|_{H_0^1(\Omega)}$$

Hence, we satisfy the assumptions in required in Theorem 3 and therefore the variational problem defined in (1) has a unique solution. $\qquad \square$

## B  Missing Proofs for Section 5

**Corollary 2.** *If $u_0$ is initialized to be identically $0$, then $u_0 \in H_0^1(\Omega)$ (as it satisfies the boundary condition), then the number of parameters required in $u_\epsilon$ is bounded by*

$$O\left( d^{2 \frac{\log\left( \frac{\|f\|_{L^2(\Omega)}}{\lambda_1 \epsilon} \right)}{\log \kappa}} (N_0 + N_A) + \frac{\log\left( \frac{\|f\|_{L^2(\Omega)}}{\lambda_1 \epsilon} \right)}{\log \kappa} (N_f + N_c) \right)$$

*Proof.* Given that $u_0$ is identically $0$, the value of $R$ in Theorem 1 equals $\|u^\star - u_0\|_{L^2(\Omega)} = \|u^\star\|_{L^2(\Omega)}$ Using the inequality in (2), we have,

$$
\begin{aligned}
\|u^\star\|_{L^2(\Omega)}^2 &\le \frac{\langle Lu^\star, u^\star \rangle}{\lambda_1} \\
&\le \frac{1}{\lambda_1} \langle f, u^\star \rangle_{L^2(\Omega)} \\
&\le \frac{1}{\lambda_1} \|f\|_{L^2(\Omega)} \|u^\star\|_{L^2(\Omega)} \\
\implies \|u^\star\|_{L^2(\Omega)} &\le \frac{1}{\lambda_1} \|f\|_{L^2(\Omega)} \qquad\qquad \square
\end{aligned}
$$

## C  Missing Proofs for Section 6

### C.1  Proof for Lemma 1

*Proof.* Writing $u \in \Phi_k$ as $u = \sum_i d_i \varphi_i$ where $d_i = \langle u, \varphi_i \rangle_{L^2(\Omega)}$, we have $Lu = \sum_{i=1}^k \lambda_i d_i \varphi_i$ Therefore $Lu \in \tilde{\Phi}_k$ and $Lu$ lies in $H_0^1(\Omega)$, proving (1).

Since $v \in \Phi_k$, we use the definition of eigenvalues in (3) to get,

$$
\begin{aligned}
\frac{\langle Lv, v \rangle_{L^2(\Omega)}}{\|v\|_{L^2(\Omega)}} &\le \sup_v \frac{\langle Lv, v \rangle_{L^2(\Omega)}}{\|v\|_{L^2(\Omega)}} = \lambda_k \\
\implies \langle Lv, v \rangle_{L^2(\Omega)} &\le \lambda_k \|v\|_{L^2(\Omega)}^2
\end{aligned}
$$

and similarly

$$
\begin{aligned}
\frac{\langle Lv, v \rangle_{L^2(\Omega)}}{\|v\|_{L^2(\Omega)}} &\ge \inf_v \frac{\langle Lv, v \rangle_{L^2(\Omega)}}{\|v\|_{L^2(\Omega)}} = \lambda_1 \\
\implies \langle Lv, v \rangle_{L^2(\Omega)} &\ge \lambda_1 \|v\|_{L^2(\Omega)}^2
\end{aligned}
$$

In order to prove (2.) let us first denote $\bar{L} := \left( I - \frac{2}{\lambda_k + \lambda_1} L \right)$. Note if $\varphi$ is an eigenfunction of $L$ with corresponding eigenvalue $\lambda$, it is also an eigenfunction of $\bar{L}$ with corresponding eigenvalue $\frac{\lambda_k + \lambda_1 - 2\lambda}{\lambda_k + \lambda_1}$.

Hence, writing $u \in \Phi_k$ as $u = \sum_{i=1}^k d_i \varphi_i$, where $d_i = \langle u, \varphi_i \rangle$, we have

$$\|\bar{L}u\|_{L^2(\Omega)}^2 = \left\| \sum_{i=1}^k \frac{\lambda_k + \lambda_1 - 2\lambda_i}{\lambda_k + \lambda_1} d_i \varphi_i \right\|_{L^2(\Omega)}^2 \le \max_{i \in k} \left( \frac{\lambda_k + \lambda_1 - 2\lambda_i}{\lambda_k + \lambda_1} \right)^2 \left\| \sum_{i=1}^k d_i \varphi_i \right\|_{L^2(\Omega)}^2 \tag{12}$$

By the orthogonality of $\{\varphi_i\}_{i=1}^k$, we have

$$\left\|\sum_{i=1}^k d_i\varphi_i\right\|_{L^2(\Omega)}^2 = \sum_{i=1}^k d_i^2 = \|u\|_{L^2(\Omega)}^2$$

Since $\lambda_1 \le \lambda_2 \cdots \le \lambda_k$, we have $\lambda_k + \lambda_1 - 2\lambda_i \ge \lambda_1 - \lambda_k$ and $\lambda_k + \lambda_1 - 2\lambda_i \le \lambda_k - \lambda_1$, so $|\lambda_k + \lambda_1 - 2\lambda_i| \le \lambda_k - \lambda_1$. This implies $\max_{i \in k}\left(\frac{\lambda_k+\lambda_1-2\lambda_i}{\lambda_k+\lambda_1}\right)^2 \le \left(\frac{\lambda_1-\lambda_k}{\lambda_1+\lambda_k}\right)^2$. Plugging this back in (12), we get the claim we wanted. $\qquad\square$

## C.2 Proof of Lemma 3

*Proof.* Given that $u_0 \in H_0^1(\Omega)$ and $u_0 \in \tilde{\Phi}_k$ the function $\tilde{L}u_0 \in H_0^1(\Omega)$ and $\tilde{L}u_0 \in \tilde{\Phi}_k$ as well (from Lemma 1).

As $\tilde{f}_{\mathrm{span}} \in \tilde{\Phi}_k$, all the iterates in the sequence will also belong to $H_0^1(\Omega)$ and will lie in the $\tilde{\Phi}_k$.

Now at a step $t$ the iteration looks like,

$$u_{t+1} = u_n - \frac{2}{\tilde{\lambda}_k + \tilde{\lambda}_1}\left(\tilde{L}u_t - \tilde{f}_{\mathrm{span}}\right)$$

$$u_{t+1} - \tilde{u}_{\mathrm{span}}^\star = \left(I - \frac{2}{\tilde{\lambda}_k + \tilde{\lambda}_1}\tilde{L}\right)(u_t - \tilde{u}_{\mathrm{span}}^\star)$$

Using the result from Lemma 1, part 3. we have,

$$\|u_{t+1} - \tilde{u}_{\mathrm{span}}^\star\|_{L^2(\Omega)} \le \left(\frac{\tilde{\lambda}_k - \tilde{\lambda}_1}{\tilde{\lambda}_k + \tilde{\lambda}_1}\right)\|u_t - \tilde{u}_{\mathrm{span}}^\star\|_{L^2(\Omega)}$$

$$\implies \|u_{t+1} - \tilde{u}_{\mathrm{span}}^\star\|_{L^2(\Omega)} \le \left(\frac{\tilde{\lambda}_k - \tilde{\lambda}_1}{\tilde{\lambda}_k + \tilde{\lambda}_1}\right)^t\|u_0 - \tilde{u}_{\mathrm{span}}^\star\|_{L^2(\Omega)}$$

Hence this implies that, $\|u_T - \tilde{u}_{\mathrm{span}}^\star\|_{L^2(\Omega)} \le \epsilon$ when

$$T \ge \frac{\log\left(\frac{\|u_0-\tilde{u}_{\mathrm{span}}^\star\|_{L^2(\Omega)}}{\epsilon}\right)}{\log\left(\frac{\tilde{\lambda}_k+\tilde{\lambda}_1}{\tilde{\lambda}_k-\tilde{\lambda}_1}\right)}$$

Using $\kappa := \frac{\tilde{\lambda}_k+\tilde{\lambda}_1}{\tilde{\lambda}_k-\tilde{\lambda}_1}$ we can rewrite the above as

$$T \ge \frac{\log\left(\frac{\|u_0-\tilde{u}_{\mathrm{span}}^\star\|_{L^2(\Omega)}}{\epsilon}\right)}{\log(\kappa)}$$

$\qquad\square$

## C.3 Important Lemmas for Section 6.2

### Operations on Neural Network functionals

**Lemma 7** (Backpropagation [36]). *Consider neural network $g : \mathbb{R}^m \to \mathbb{R}$ with depth $l$, $N$ parameters and differentiable activation functions in the set $\{\sigma_i\}_{i=1}^A$. There exists a neural network of size $O(l + N)$ and activation functions in the set $\{\sigma_i, \sigma_i'\}_{i=1}^A$ that calculates the gradient $\frac{dg}{di}$ for all $i \in [m]$.*

**Lemma 8** (Addition and Multiplication). *Given neural networks $g : \Omega \to \mathbb{R}$, $h : \Omega \to \mathbb{R}$, with $N_g$ and $N_h$ parameters respectively, the operations $g(x) + h(x)$ and $g(x) \cdot h(x)$ can be represented by neural networks of size $O(N_g + N_h)$, and square activation functions.*

*Proof.* For *Addition*, there exists a network $h$ containing both networks $f$ and $g$ as subnetworks and an extra layer to compute the addition between their outputs. Hence, the total number of parameters in such a network will be $O(N_f + N_g)$.

For *Multiplication*, consider the operation $f(x) \cdot g(x) = \frac{1}{2}\left((f(x) + g(x))^2 - f(x)^2 - g(x)^2\right)$. Then following the same argument as for addition of two networks, we can construct a network $h$ containing both networks and square activation function. $\square$

While the representation result in Lemma 8 is shown using square activation, we refer to [41] for approximation results with ReLU activation. The scaling with respect to the number of parameters in the network remains the same.

### C.4 Remarks About Activation Functions

In this section, we make some remarks about the activations used in our theorem statements. Namely, we show that using standard techniques from approximation theory [18, 41], one can approximate a neural network with one choice of nonlinearity via a (comparably sized) neural network with another choice of nonlinearity, under very mild conditions on the nonlinearities. Crucially, this simulation only increases the size by a dimension-independent factor. This result frees us (for purposes of deriving an expressibility result) to work with activation functions chosen for mathematical convenience and produce results that hold without loss of generality.

We present the following lemma for ReLU activation function however we note that the proofs for other activations like sigmoid or tanh can be written completely analogously). We note that this proof is almost verbatim the same as the proof of Lemma 1.3 in Telgarsky [38].

**Lemma 9.** *Let $\Omega \subseteq [-M, M]^d$ and let $G_1 : [-M, M]^d \to \mathbb{R}$ be a neural network with at most $l$ layers and $n$ parameters, such that the weights $W^{(i)}$ for each layer $i$ and node $j$ in $G_1$ are bounded, i.e, for all $i, j$ we have that $\sum_k |W^{(i)}_{jk}| \leq B$. Furthermore, assume that the activation functions used in $G_1$ belong to the set $\Xi$, such for all $\sigma : \mathbb{R} \to \mathbb{R}$, $\sigma \in \Xi$ we have that $\sup_{x \in [-B \cdot M, B \cdot M]} \sigma \leq M$ and the Lipschitz constant $L$. Then there exists a neural network $G_2$ with ReLU activation and $O\left(\frac{(LB)^l M}{\epsilon'} \log(\frac{(LB)^l M}{\epsilon'})\right)$ parameters we have $\sup_{x \in [-M, M]^d} |G_1(x) - G_2(x)| \leq \epsilon$.*

*Proof.* For any $\sigma \in \Xi$ from Theorem 1 in Yarotsky [41] it follows that there exists a neural network $R$ with ReLU activations, and $O\left(\frac{LBM}{\epsilon'} \log\left(\frac{LBM}{\epsilon'}\right)\right)$ parameters such that $\sup_{x \in [-B \cdot M, B \cdot M]} |\sigma(x) - R(x)| \leq \epsilon'$.

Now we will construct the network $G_2$ by replacing each activation in $G_1$ with the corresponding network $R$ as given by the result above with $\epsilon' = \epsilon/l$.

Note, this network is at most a factor of $O\left(\frac{(LB)^l M}{\epsilon'} \log(\frac{(LB)^l M}{\epsilon'})\right)$ bigger than $G_1$, as the lemma requires.

We will prove the claim of the lemma by induction on $l$. More precisely, we will show (by induction) that for each node at layer $i$, the network $G_2$ calculates a function that is $(LB)^i i\epsilon'$ away in $l_\infty$ norm from the corresponding node in $G_1$, and the inputs to the node are in $[-BM, BM]$.

For the base case $i = 1$, since the input $x \in [-M, M]^d$, the result follows by Theorem 1 in Yarotsky [41].

We proceed to the inductive claim. Let $H(x)$ denote the vector valued mapping computed by the nodes at layer $i$, and let $H_R(x)$ be the corresponding vector in $G_2$. As inductive hypothesis, we assume that $\|H(x) - H_R(x)\|_\infty \leq (LB)^i i\epsilon'$ for all $x \in [-M, M]^d$ and $\|H(x)\|_\infty \leq M$ as well as $\|H_R(x)\|_\infty \leq M$.

Therefore, for the $j^{th}$ node in layer $(i+1)$ in network $G_1$ we have $|W_j^T H(x)| \leq \|W_j\|_1 \|H(x)\|_\infty \leq BM$ and $\sigma$ is bounded by $M$ on this interval, so we have $\|\sigma_1(W_j^T H(x))\|_\infty \leq M$. Along with the bound on the activations, the part of the inductive hypothesis about the size of the input in proven.

To prove the error bound, we have:

$$|\sigma(W_j^T H(x)) - R(W_j^T H_R(x))| \leq |\sigma(W_j^T H(x)) - \sigma(W_j^T H_R(x))| + |\sigma(W_j^T H_R(x)) - R(W_j^T H_R(x))|$$
$$\leq L|W_j^T(H(x) - H_R(x))| + \epsilon'$$
$$\leq L\|W_j\|_1 \|H(x) - H_R(x)\|_\infty + \epsilon'$$
$$\leq (LB)^{i+1}(i+1)\epsilon'$$

This finishes the proof of the inductive step, and thus the lemma. $\qquad\square$

Therefore, Lemma 9 can be used to approximate the network $u_\epsilon$ defined in Theorem 1 with a network $v_\epsilon$ that uses ReLU activation *without* a worse dependence on the dimension, though there will be dependence on other quantities like the weights in the PDE coefficient networks (precisely, the maximum sum of weights coming in and going out of a node in the network), the maximum depth of these networks and their Lipschitz constants.

Moreover, these quantities can be bounded for the network produced in the proof of Theorem 1. The main reason for this is that all the operations in our proof (addition, multiplication and backpropagation through Lemma 7) do not create nodes with weights into and out of a node bigger than the original network. Precisely:

**Corollary 3.** *Assume that the maximum depth of the neural networks $\tilde{A}, \tilde{c}$ and $u_0$ is l with activation functions in the set $\Xi$ (as defined in Lemma 9). Furthermore, that assume that for each network each layer i and node j satisfies: $\sum_k |W_{j,k}^{(i)}| \leq B$ and $\sum_k |W_{k,j}^{(i)}| \leq B$ for $B \geq 2$ (i.e, the "in-weights" and "out-weights" of each node are bounded by B). With $\tilde{\epsilon}, \epsilon$ and $T$ defined as in Theorem 1, there exists a neural network $v_\epsilon$, such that,*

- *$v_\epsilon$ uses ReLU activations only*

- *$\|v_\epsilon - u_\epsilon\|_\infty \leq \epsilon$ and,*

- *$v_\epsilon$ has $O\left(N_T \frac{(LB)^D DBM}{\epsilon} \log\left(\frac{(LB)^D DBM}{\epsilon}\right)\right)$ parameters where $D = O(c^T l)$ and $N_T = d^{2T}(N_0 + N_A + T(N_f + N_c))$ where $c \leq 5$ is a constant. (Note, here $N_T$ is the size bound obtained in Theorem 1)*

*Proof.* First, we show the following: (i) the network $u_\epsilon$ satisfies $\forall i, j : \sum_k |W_{j,k}^{(i)}| \leq B$ and (ii) has depth bounded by $D = O(c^T l)$.

To show (i), we will show that each of the operations we employ (addition, multiplication, taking derivatives) maintains this condition. Notice that multiplication and addition each *add* one node, with 2 incoming weights bounded by 1. Since $B \geq 2$, the claim obtains for these operations. Continuing to differentiation, the construction in Lemma 7 constructs a network that has two copies of each of nodes in the original network: one for the "forward" network, and one for the "backward" network (in our notation, the latter nodes are $\frac{\partial h_s}{\partial g_s}$). The first types of nodes have exactly the same children as the original network, so for those nodes $v$ we have $\sum_{k \in \text{child of } v} |W_{v,k}| \leq B$. On the other hand, for the latter kinds of nodes, the children of the node are the *parents* of the node in the original network. Since in our assumptions, we also assumed $\forall i, j : \sum_k |W_{k,j}^{(i)}| \leq B$, for these nodes too we have $\sum_{k \in \text{child of } v} |W_{v,k}| \leq B$. Thus, differentiation also maintains the bound $B$, proving (i).

Now, we can apply the lemma from the previous reply to produce a network $v_\epsilon$ that has size $O(N_T \frac{(LB)^D DBM}{\epsilon} \log(\frac{(LB)^D DBM}{\epsilon}))$ where $D = O(c^T l)$ and $N_T = d^{2T}(N_0 + N_A + T(N_f + N_c))$. $\qquad\square$

# D Perturbation Analysis

## D.1 Proof of Lemma 2

*Proof.* Using the triangle inequality the error between $u^\star$ and $\tilde{u}^\star_{\mathrm{span}}$, we have,

$$\|u^\star - \tilde{u}^\star_{\mathrm{span}}\|_{L^2(\Omega)} \leq \underbrace{\|u^\star - u^\star_{\mathrm{span}}\|_{L^2(\Omega)}}_{(I)} + \underbrace{\|u^\star_{\mathrm{span}} - \tilde{u}^\star_{\mathrm{span}}\|_{L^2(\Omega)}}_{(II)} \tag{13}$$

where $u^\star_{\mathrm{span}}$ is the solution to the PDE $Lu = f_{\mathrm{span}}$.

In order to bound Term (I), we use the inequality in (2) to get,

$$\|u^\star - u^\star_{\mathrm{span}}\|^2_{L^2(\Omega)} \leq \frac{1}{\lambda_1}\langle L(u^\star - u^\star_{\mathrm{span}}), u^\star - u^\star_{\mathrm{span}}\rangle_{L^2(\Omega)}$$

$$= \frac{1}{\lambda_1}\langle f - f_{\mathrm{span}}, u^\star - u^\star_{\mathrm{span}}\rangle_{L^2(\Omega)}$$

$$\leq \frac{1}{\lambda_1}\|f - f_{\mathrm{span}}\|_{L^2(\Omega)}\|u^\star - u^\star_{\mathrm{span}}\|_{L^2(\Omega)}$$

$$\implies \|u^\star - u^\star_{\mathrm{span}}\|_{L^2(\Omega)} \leq \frac{1}{\lambda_1}\|f - f_{\mathrm{span}}\|_{L^2(\Omega)} \leq \frac{\epsilon_{\mathrm{span}}}{\lambda_1} \tag{14}$$

We now bound Term (II).

First we introduce an intermediate PDE $Lu = \tilde{f}_{\mathrm{span}}$, and denote the solution $\tilde{u}$. Therefore, by utilizing triangle inequality again Term (II) can be expanded as the following,

$$\|u^\star_{\mathrm{span}} - \tilde{u}^\star_{\mathrm{span}}\|_{L^2(\Omega)} \leq \|u^\star_{\mathrm{span}} - \tilde{u}\|_{L^2(\Omega)} + \|\tilde{u} - \tilde{u}^\star_{\mathrm{span}}\|_{L^2(\Omega)} \tag{15}$$

We will tackle the second term in (15) first.

Using $\tilde{u} = L^{-1}\tilde{f}_{\mathrm{span}}$ and $\tilde{u}^\star_{\mathrm{span}} = \tilde{L}^{-1}\tilde{f}_{\mathrm{span}}$,

$$\|\tilde{u} - \tilde{u}^\star_{\mathrm{span}}\|_{L^2(\Omega)} = \|(L^{-1} - \tilde{L}^{-1})\tilde{f}_{\mathrm{span}}\|_{L^2(\Omega)}$$

$$= \|(L^{-1}\tilde{L} - I)\tilde{L}^{-1}\tilde{f}_{\mathrm{span}}\|_{L^2(\Omega)}$$

$$\implies \|\tilde{u} - \tilde{u}^\star_{\mathrm{span}}\|_{L^2(\Omega)} = \|(L^{-1}\tilde{L} - I)\tilde{u}^\star_{\mathrm{span}}\|_{L^2(\Omega)} \tag{16}$$

Further, using (35) from Lemma 13, we have for all $u \in H^1_0(\Omega)$,

$$\langle(\tilde{L} - L)u, u\rangle_{L^2(\Omega)} \leq \delta\langle Lu, u\rangle_{L^2(\Omega)}$$

$$\implies \langle(\tilde{L}L^{-1} - I)Lu, u\rangle_{L^2(\Omega)} \leq \delta\langle Lu, u\rangle_{L^2(\Omega)}$$

$$\implies \langle(\tilde{L}L^{-1} - I)v, u\rangle_{L^2(\Omega)} \leq \delta\langle v, u\rangle_{L^2(\Omega)}$$

$$\implies \langle(\tilde{L}L^{-1})v, u\rangle_{L^2(\Omega)} \leq (1 + \delta)\langle v, u\rangle_{L^2(\Omega)} \tag{17}$$

where $v = Lu$. Therefore using (17) the following holds for all $u \in H^1_0(\Omega)$,

$$\langle(\tilde{L}L^{-1})u, u\rangle_{L^2(\Omega)} \leq (1 + \delta)\|u\|^2_{L^2(\Omega)} \tag{18}$$

$$\overset{(1)}{\implies} \langle u, (L^{-1}\tilde{L})u\rangle_{L^2(\Omega)} \leq (1 + \delta)\|u\|^2_{L^2(\Omega)}$$

$$\overset{(2)}{\implies} \langle(L^{-1}\tilde{L} - I)u, u\rangle_{L^2(\Omega)} \leq \delta\|u\|^2_{L^2(\Omega)} \tag{19}$$

where we use the fact that the operators $t\tilde{L}$ and $L^{-1}$ are self-adjoint to get (1) and then bring the appropriate terms to the LHS in (2). Therefore, using the inequality in (19) and inequality in (3) (with $L^{-1}L$ as the operator), we can upper bounded (16) to get,

$$\|\tilde{u} - \tilde{u}^\star_{\mathrm{span}}\|_{L^2(\Omega)} \leq \delta\|\tilde{u}^\star_{\mathrm{span}}\|_{L^2(\Omega)} \tag{20}$$

where $\delta = \frac{1}{\min\{m/\epsilon_A, \zeta/\epsilon_c\}}$.

Proceeding to the first term in (15), using Lemma 11, and the inequality in (2), the term $\|u^\star_{\text{span}} - \tilde{u}\|_{L^2(\Omega)}$ can be upper bounded by,

$$
\begin{aligned}
\|u^\star_{\text{span}} - \tilde{u}\|^2_{L^2(\Omega)} &\leq \frac{1}{\lambda_1} \langle L(u^\star_{\text{span}} - \tilde{u}), u^\star_{\text{span}} - \tilde{u} \rangle_{L^2(\Omega)} \\
&\leq \frac{1}{\lambda_1} \langle f_{\text{span}} - \tilde{f}_{\text{span}}, u^\star_{\text{span}} - \tilde{u} \rangle_{L^2(\Omega)} \\
&\leq \frac{1}{\lambda_1} \|f_{\text{span}} - \tilde{f}_{\text{span}}\|_{L^2(\Omega)} \|u^\star_{\text{span}} - \tilde{u}\|_{L^2(\Omega)} \\
\implies \|u^\star_{\text{span}} - \tilde{u}\|_{L^2(\Omega)} &\leq \frac{1}{\lambda_1} \|f_{\text{span}} - \tilde{f}_{\text{span}}\|_{L^2(\Omega)} \leq \frac{\delta}{\lambda_1} \cdot \frac{2^{3/2} \|f\|_{L^2(\Omega)}}{\gamma - \delta}
\end{aligned}
\tag{21}
$$

Therefore Term (II), i.e., $\|u^\star_{\text{span}} - \tilde{u}^\star_{\text{span}}\|_{L^2(\Omega)}$ can be upper bounded by

$$
\|u^\star_{\text{span}} - \tilde{u}^\star_{\text{span}}\|_{L^2(\Omega)} \leq \|u^\star_{\text{span}} - \tilde{u}\|_{L^2(\Omega)} + \|\tilde{u} - \tilde{u}^\star_{\text{span}}\|_{L^2(\Omega)} \leq \frac{\hat{\epsilon}_f}{\lambda_1} + \delta \|\tilde{u}^\star_{\text{span}}\|_{L^2(\Omega)} \tag{22}
$$

Putting everything together, we can upper bound (13) as

$$
\begin{aligned}
\|u^\star - \tilde{u}^\star_{\text{span}}\|_{L^2(\Omega)} &\leq \|u^\star - u^\star_{\text{span}}\|_{L^2(\Omega)} + \|u^\star_{\text{span}} - \tilde{u}^\star_{\text{span}}\|_{L^2(\Omega)} \\
&\leq \frac{\epsilon_{\text{span}}}{\lambda_1} + \frac{\delta}{\lambda_1} \frac{2^{3/2} \|f\|_{L^2(\Omega)}}{\gamma - \delta} + \delta \|\tilde{u}^\star_{\text{span}}\|_{L^2(\Omega)}
\end{aligned}
$$

where $\gamma = \frac{1}{\lambda_k} - \frac{1}{\lambda_{k+1}}$ and $\delta = \frac{1}{\min\{m/\epsilon_A, \zeta/\epsilon_c\}}$. $\qquad\square$

### D.2 Proof of Lemma 5

*Proof.* We define $r = \tilde{f}_{\text{span}} - f_{\text{nn}}$, therefore from Lemma 12 we have that for any multi-index $\alpha$,

$$
\|\tilde{L}^{(t)} r\|_{L^2(\Omega)} \leq (t!)^2 \cdot C^t (\epsilon_{\text{nn}} + \epsilon_{\text{span}}) + \lambda_k^t \frac{\|f_{\text{span}}\|_{L^2(\Omega)} 2^{3/2} \delta}{\gamma - \delta}.
$$

For every $t \in \mathbb{N}$, we will write $u_t = \hat{u}_t + r_t$, s.t. $\hat{u}_t$ is a neural network and we (iteratively) bound $\|r_t\|_{L^2(\Omega)}$. Precisely, we define a sequence of neural networks $\{\hat{u}_t\}_{t=0}^\infty$, s.t.

$$
\begin{cases}
\hat{u}_0 = u_0, \\
\hat{u}_{t+1} = \hat{u}_t - \eta \left( \tilde{L} \hat{u}_t - f_{\text{nn}} \right)
\end{cases}
$$

Since $r_t = u_t - \hat{u}_t$, we can define a corresponding recurrence for $r_t$:

$$
\begin{cases}
r_0 = 0, \\
r_{t+1} = (I - \eta \tilde{L}) r_t - r
\end{cases}
$$

Unfolding the recurrence, we get

$$
r_{t+1} = \sum_{i=0}^{t} (I - \eta \tilde{L})^{(i)} r \tag{23}
$$

Using the binomial expansion we can write:

$$(I - \eta \tilde{L})^{(t)} r = \sum_{i=0}^{(t)} \binom{t}{i} (-1)^i (\eta \tilde{L})^{(i)} r$$

$$\implies \|(I - \eta \tilde{L})^{(t)} r\|_{L^2(\Omega)} = \left\| \sum_{i=0}^{t} \binom{t}{i} (-1)^i (\eta \tilde{L})^{(i)} r \right\|_{L^2(\Omega)}$$

$$\leq \sum_{i=0}^{t} \binom{t}{i} \eta^i \|\tilde{L}^{(i)} r\|_{L^2(\Omega)}$$

$$\leq \sum_{i=0}^{t} \left(\frac{te}{i}\right)^i \eta^i \|\tilde{L}^{(i)} r\|_{L^2(\Omega)} \qquad \because \binom{t}{i} \leq \left(\frac{te}{i}\right)^i$$

$$\leq \sum_{i=0}^{t} \left(\frac{te}{i}\eta\right)^i \left( (i!)^2 C^i \left(\epsilon_{\mathrm{nn}} + \epsilon_{\mathrm{span}}\right) + \lambda_k^i \frac{\|f_{\mathrm{span}}\|_{L^2(\Omega)} 2^{3/2}\delta}{\gamma - \delta} \right) \quad \text{from Lemma 12}$$

$$\leq \sum_{i=0}^{t} \left(\frac{te}{i}\eta\right)^i (i!)^2 C^i \left( \left(\epsilon_{\mathrm{nn}} + \epsilon_{\mathrm{span}}\right) + \frac{\lambda_k^i}{(i! C^i)} \frac{\|f_{\mathrm{span}}\|_{L^2(\Omega)} 2^{3/2}\delta}{\gamma - \delta} \right)$$

$$\leq \sum_{i=0}^{t} \left(\frac{te}{i}\eta i^2 C\right)^i \left( \left(\epsilon_{\mathrm{nn}} + \epsilon_{\mathrm{span}}\right) + \frac{\lambda_k^i}{(i!)^2 C^i} \frac{\|f_{\mathrm{span}}\|_{L^2(\Omega)} 2^{3/2}\delta}{\gamma - \delta} \right) \quad \because i! \leq i^i$$

$$\leq \sum_{i=0}^{t} \left(\frac{te}{i}\eta i^2 C\right)^i \left( \left(\epsilon_{\mathrm{nn}} + \epsilon_{\mathrm{span}}\right) + \lambda_k^i \frac{\|f_{\mathrm{span}}\|_{L^2(\Omega)} 2^{3/2}\delta}{\gamma - \delta} \right) \quad \because \frac{1}{(i!)^2 C^i} \leq 1$$

$$\leq \sum_{i=0}^{t} (tie\eta C)^i \left( \left(\epsilon_{\mathrm{nn}} + \epsilon_{\mathrm{span}}\right) + \lambda_k^i \frac{\|f_{\mathrm{span}}\|_{L^2(\Omega)} 2^{3/2}\delta}{\gamma - \delta} \right)$$

$$\leq t(t^2 e\eta C)^t \left( \left(\epsilon_{\mathrm{nn}} + \epsilon_{\mathrm{span}}\right) + \lambda_k^t \frac{\|f_{\mathrm{span}}\|_{L^2(\Omega)} 2^{3/2}\delta}{\gamma - \delta} \right)$$

$$\leq t(t^2 e\eta C)^t \left( \epsilon_{\mathrm{nn}} + \epsilon_{\mathrm{span}} + \lambda_k^t \frac{\|f_{\mathrm{span}}\|_{L^2(\Omega)} 2^{3/2}\delta}{\gamma - \delta} \right)$$

Hence,

$$\|r_t\|_{L^2(\Omega)} \leq t^2 \max\{1, (t^2 e\eta C)^t\} \left( \epsilon_{\mathrm{nn}} + \epsilon_{\mathrm{span}} + \lambda_k^t \frac{\|f_{\mathrm{span}}\|_{L^2(\Omega)} 2^{3/2}\delta}{\gamma - \delta} \right)$$

$\square$

# E   Technical Lemmas: Perturbation Bounds

In this section we introduce some useful lemmas about perturbation bounds used in the preceding parts of the appendix.

First we show a lemma that's ostensibly an application of Davis-Kahan to the (bounded) operators $L^{-1}$ and $\tilde{L}^{-1}$:

**Lemma 10** (Subspace alignment). *Consider linear elliptic operators $L$ and $\tilde{L}$ with eigenvalues $\lambda_1 \leq \lambda_2 \leq \cdots$ and $\lambda_1 \leq \lambda_2 \leq \cdots$ respectively. Assume that $\gamma := \frac{1}{\lambda_k} - \frac{1}{\lambda_{k+1}} > 0$. Then, there exists an orthogonal transformation $O : H_0^1(\Omega) \to H_0^1(\Omega)$ such that the first $k$ eigenfunctions of $L$ and $\tilde{L}$ satisfy,*

$$\sup_{\substack{a \in \mathbb{R}^k \\ \sum_{i=1}^{k} a_i^2 = 1}} \left\| \sum_{i=1}^{k} (O\tilde{\varphi}_i - \varphi_i) a_i \right\|_{L^2(\Omega)} \leq \frac{2^{3/2}\delta}{\gamma - \delta}$$

*where $\delta = \frac{1}{\min\{m/\epsilon_A, \zeta/\epsilon_c\}}$.*

*Proof.* From (19), with $\delta = \frac{1}{\min\{m/\epsilon_A, \zeta/\epsilon_c\}}$ we know the following,

$$\langle (L^{-1}\tilde{L} - I)u, u \rangle_{L^2(\Omega)} \leq \delta \|u\|^2_{L^2(\Omega)}$$

$$\implies \langle (L^{-1} - \tilde{L}^{-1})\tilde{L}u, u \rangle_{L^2(\Omega)} \leq \delta \|u\|^2_{L^2(\Omega)}$$

$$\implies \langle (L^{-1} - \tilde{L}^{-1})v, u \rangle_{L^2(\Omega)} \leq \delta \|u\|^2_{L^2(\Omega)}$$

Now, the operator norm $\|L^{-1} - \tilde{L}^{-1}\|$ can be written as,

$$\|L^{-1} - \tilde{L}^{-1}\| = \sup_{v \in H_0^1(\Omega)} \frac{\langle (L^{-1} - \tilde{L}^{-1})v, v \rangle_{L^2(\Omega)}}{\|v\|^2_{L^2(\Omega)}} \leq \delta \tag{24}$$

Further note that, $\{\frac{1}{\lambda_i}\}_{i=1}^\infty$ and $\{\frac{1}{\tilde{\lambda}_i}\}_{i=1}^\infty$ are the eigenvalues of the operators $L^{-1}$ and $\tilde{L}^{-1}$, respectively. Therefore from *Weyl's Inequality* and (24) we have

$$\sup_i \left| \frac{1}{\lambda_i} - \frac{1}{\tilde{\lambda}_i} \right| \leq \|L^{-1} - \tilde{L}^{-1}\| \leq \delta \tag{25}$$

Therefore, for all $i \in \mathbb{N}$, we have that $\frac{1}{\tilde{\lambda}_i} \in [\frac{1}{\lambda_i} - \delta, \frac{1}{\lambda_i} + \delta]$, i.e., all the eigenvalues of $\tilde{L}^{-1}$ are within $\delta$ of the eigenvalue of $L^{-1}$. which therefore implies that the difference between $k^{th}$ eigenvalues is,

$$\frac{1}{\tilde{\lambda}_k} - \frac{1}{\lambda_{k+1}} \geq \frac{1}{\lambda_k} - \frac{1}{\lambda_{k+1}} - \delta$$

Since the operators $L^{-1}, \tilde{L}^{-1}$ are bounded, Davis Kahan [8] can be applied to conclude that for all $x \in \Omega$,

$$\|\sin\Theta(V, \tilde{V})\| \leq \frac{\|L^{-1} - \tilde{L}^{-1}\|}{\gamma - \delta} \leq \frac{\delta}{\gamma - \delta} \tag{26}$$

where $\|\cdot\|$ is understood to be the operator norm, $V = \Phi_k$ and $\tilde{V} = \tilde{\Phi}_k$. Via the definition of $\sin\Theta$ distance, (26) also implies that there exists an orthogonal transformation $O : H_0^1(\Omega) \to H_0^1(\Omega)$ such that

$$\sup_{\substack{a \in \mathbb{R}^k \\ \sum_{i=1}^k a_i^2 = 1}} \left\| \sum_{i=1}^k (O\tilde{\varphi}_i - \varphi_i)a_i \right\|_{L^2(\Omega)} \leq \frac{2^{3/2}\|L^{-1} - \tilde{L}^{-1}\|}{\gamma - \delta} \leq \frac{2^{3/2}\delta}{\gamma - \delta} \tag{27}$$

$\square$

In the next lemma, we use the result in Lemma 10 to show that the difference between $f_{\text{span}}$ and $\tilde{f}_{\text{span}}$ is small.

**Lemma 11** (Bounding distance between $f_{\text{span}}$ and $\tilde{f}_{\text{span}}$)**.** *Given Assumptions (i)-(iii)and $f_{\text{span}} \in \Phi_k$, there exists a function $\tilde{f}_{\text{span}} \in \tilde{\Phi}_k$ s.t.*

$$\|f_{\text{span}} - \tilde{f}_{\text{span}}\|_{L^2(\Omega)} \leq \frac{\|f_{\text{span}}\|_{L^2(\Omega)} 2^{3/2}\delta}{\gamma - \delta} \tag{28}$$

*where $\delta = \frac{1}{\min\{m/\epsilon_A, \zeta/\epsilon_c\}}$.*

*Proof.* Let us write $f_{\text{span}} = \sum_{i=1}^k f_i \varphi_i$ where $f_i = \langle f_{\text{span}}, \varphi_i \rangle_{L^2(\Omega)}$. Further, we can define a function $\tilde{f}_{\text{span}} \in \tilde{\Phi}_k$ such that $\tilde{f}_{\text{span}} = \sum_{i=1}^k f_i O\tilde{\varphi}_i$. Using the result in (27), and we have

$$\|f_{\text{span}} - \tilde{f}_{\text{span}}\|_{L^2(\Omega)} = \left\| \sum_{i=1}^k f_i (\varphi_i - O\tilde{\varphi}_i) \right\|_{L^2(\Omega)}$$

$$\leq \|f_{\text{span}}\|_{L^2(\Omega)} \sup_{\substack{a \in \mathbb{R}^k \\ \sum_{i=1}^k a_i^2 = 1}} \left\| \sum_{i=1}^k (O\tilde{\varphi}_i - \varphi_i)a_i \right\|_{L^2(\Omega)}$$

$$\leq \|f_{\text{span}}\|_{L^2(\Omega)} \frac{2^{3/2}\delta}{\gamma - \delta}$$

where $\gamma = \frac{1}{\lambda_k} - \frac{1}{\lambda_{k+1}}$, and $\delta = \frac{1}{\min\{m/\epsilon_A, \zeta/\epsilon_c\}}$.

$\square$

Finally, we show that repeated applications of $\tilde{L}$ to $f_{nn} - f$ have also bounded norms:

**Lemma 12** (Bounding norms of applications of $\tilde{L}$). *The functions $f_{nn}$ and $f$ satisfy:*

1. $\|\tilde{L}^{(n)}(f_{nn} - f_{\text{span}})\|_{L^2(\Omega)} \leq (n!)^2 \cdot C^n (\epsilon_{\text{span}} + \epsilon_{nn})$

2. $\|\tilde{L}^{(n)}(f_{nn} - \tilde{f}_{\text{span}})\|_{L^2(\Omega)} \leq (n!)^2 \cdot C^n (\epsilon_{\text{span}} + \epsilon_{nn}) + \lambda_k^n \frac{\|f_{\text{span}}\|_{L^2(\Omega)} 2^{3/2} \delta}{\gamma - \delta}$

*where $\delta = \frac{1}{\min\{m/\epsilon_A, \zeta/\epsilon_c\}}$.*

*Proof.* For Part 1, by Lemma 16 we have that

$$\|\tilde{L}^{(n)}(f_{nn} - f_{\text{span}})\|_{L^2(\Omega)} \leq (n!)^2 \cdot C^n \max_{\alpha:|\alpha|\leq n+2} \|\partial^\alpha (f_{nn} - f_{\text{span}})\|_{L^2(\Omega)} \tag{29}$$

From Assumptions (i)-(iii), for any multi-index $\alpha$ we have:

$$\|\partial^\alpha f_{nn} - \partial^\alpha f_{\text{span}}\|_{L^2(\Omega)} \leq \|\partial^\alpha f_{nn} - \partial^\alpha f\|_{L^2(\Omega)} + \|\partial^\alpha f - \partial^\alpha f_{\text{span}}\|_{L^2(\Omega)}$$
$$\leq \epsilon_{nn} + \epsilon_{\text{span}} \tag{30}$$

Combining (29) and (30) we get the result for Part 1.

For Part 2 we have,

$$\|\tilde{L}^{(n)}(\tilde{f}_{\text{span}} - f_{nn})\|_{L^2(\Omega)} = \|\tilde{L}^{(n)}(\tilde{f}_{\text{span}} - f_{\text{span}} + f_{\text{span}} - f_{nn})\|_{L^2(\Omega)} \tag{31}$$
$$\leq \|\tilde{L}^{(n)}(\tilde{f}_{\text{span}} - f_{\text{span}})\|_{L^2(\Omega)} + \|\tilde{L}^{(n)}(f_{\text{span}} - f_{nn})\|_{L^2(\Omega)} \tag{32}$$

Note that from equation (18) in Lemma 10 we have that $\|L^{-1}\tilde{L} - I\| \leq \delta$ (where $\|\cdot\|$ denotes the operator norm). This implies that there exists a $\Sigma$, such that $\|\Sigma\| \leq \delta$ and we can express $\tilde{L}$ as:

$$\tilde{L} = L(I + \Sigma)$$

We will show that there exists a $\tilde{\Sigma}$, s.t. $\|\tilde{\Sigma}\| \leq n2\delta$ and $\tilde{L}^{(n)} = (I + \tilde{\Sigma})L^{(n)}$. Towards that, we will denote $L^{-(n)} := \underbrace{L^{-1} \circ L^{-1} \circ \cdots L^{-1}}_{n \text{ times}}$ and show that

$$\left\|L^{-(n)}\tilde{L}^{(n)}\right\| \leq 1 + n2\delta$$

We have:

$$\left\|L^{-(n)}\tilde{L}^{(n)}\right\| = \left\|L^{-(n)}\left(L(I+\Sigma)\right)^{(n)}\right\|$$
$$= \left\|L^{-(n)}\left(L^{(n)} + \sum_{j=1}^n L^{(j-1)} \circ (L \circ \Sigma) \circ L^{(n-j)} + \cdots + (L \circ \Sigma)^{(n)}\right)\right\|$$
$$= \left\|I + \sum_{j=1}^n L^{-(n)} \circ L^{(j-1)} \circ \Sigma \circ L^{(n-j)} + \cdots + L^{-(n)} \circ (L \circ \Sigma)^{(n)}\right\|$$
$$\leq^{(1)} 1 + \left\|\sum_{j=1}^n L^{-(n)} \circ L^{(j-1)} \circ \Sigma \circ L^{(n-j)}\right\| + \cdots + \|L^{-(n)} \circ (L \circ \Sigma)^{(n)}\|$$
$$\leq^{(2)} 1 + \sum_{i=1}^n \binom{n}{i}\delta^i$$
$$= (1 + \delta)^n$$
$$\leq^{(3)} e^{n\delta}$$
$$\leq 1 + 2n\delta$$

where (1) follows from triangle inequality, (2) follows from Lemma 17, (3) follows from $1 + x \le e^x$, and the last part follows from $n\delta \le 1/10$ and Taylor expanding $e^x$.

Next, since $L$ and $\tilde{L}$ are elliptic operators, we have $\|L^{-(n)}\tilde{L}^{(n)}\| = \|\tilde{L}^{(n)}L^{-(n)}\|$. From this, it immediately follows that there exists a $\tilde{\Sigma}$, s.t. $\tilde{L}^{(n)} = (I + \tilde{\Sigma})L^{(n)}$ with $\|\tilde{\Sigma}\| \le n2\delta$.

Plugging this into the first term of (32), we have

$$
\begin{aligned}
\|\tilde{L}^{(n)}(\tilde{f}_{\text{span}} - f_{\text{span}})\|_{L^2(\Omega)} &= \|\tilde{L}^{(n)}\tilde{f}_{\text{span}} - \tilde{L}^{(n)}f_{\text{span}}\|_{L^2(\Omega)} \\
&= \|\tilde{L}^{(n)}\tilde{f}_{\text{span}} - (I + \tilde{\Sigma})L^{(n)}f_{\text{span}}\|_{L^2(\Omega)} \\
&\le \|\tilde{L}^{(n)}\tilde{f}_{\text{span}} - L^{(n)}f_{\text{span}}\|_{L^2(\Omega)} + \|\tilde{\Sigma}L^{(n)}f_{\text{span}}\|_{L^2(\Omega)} \\
&\le \|\tilde{L}^{(n)}\tilde{f}_{\text{span}} - L^{(n)}f_{\text{span}}\|_{L^2(\Omega)} + \|\tilde{\Sigma}\|\|L^{(n)}f_{\text{span}}\|_{L^2(\Omega)} \\
&\le \|\tilde{L}^{(n)}\tilde{f}_{\text{span}} - L^{(n)}f_{\text{span}}\|_{L^2(\Omega)} + n2\delta\lambda_k^n\|f_{\text{span}}\|_{L^2(\Omega)} \qquad (33)
\end{aligned}
$$

Following Lemma 11, we know that $f_{\text{span}} = \sum_{i=1}^{k} f_i\tilde{\varphi}_i$ where $f_i = \langle f_{\text{span}}, \varphi_i\rangle$ and we define $\tilde{f}_{\text{span}} = \sum_{i=1}^{k} f_i\tilde{\varphi}_i$. Further from (25) in Lemma 10 we have for all $i \in \mathbb{N}$

$$
\left|\frac{1}{\tilde{\lambda}_i} - \frac{1}{\lambda_i}\right| \le \delta
$$

From this, we can conclude:

$$
\left|\tilde{\lambda}_i - \lambda_i\right| \le \delta\lambda_i\tilde{\lambda}_i
$$

Writing $\tilde{\lambda}_i = (1 + \tilde{e}_i)\lambda_i$ (where $\tilde{e}_i = \delta\tilde{\lambda}_i$), we have

$$
\begin{aligned}
\left|\tilde{\lambda}_i^n - \lambda_i^n\right| &= |((1 + \tilde{e}_i)\lambda_i)^n - \lambda_i^n| \\
&= |\lambda_i^n((1 + \tilde{e}_i)^n - 1)| \\
&\le^{(1)} \lambda_i^n|\tilde{e}|_i\left|\sum_{j=1}^{n}(1 + \tilde{e}_i)^j\right| \\
&\le^{(2)} \lambda_i^n n|\tilde{e}_i|e^{n|\tilde{e}_i|} \\
&\le^{(3)} \lambda_i^n n|\tilde{e}_i|(1 + |2n\tilde{e}_i|) \\
&\le 2\lambda_i^n n|\tilde{e}_i|
\end{aligned}
$$

where (1) follows from the factorization $a^n - b^n = (a - b)(\sum_{i=0}^{n-1} a^i b^{n-i-i})$, (2) follows from $1 + x \le e^x$, and (3) follows from $n|\tilde{e}_i| \le 1/10$ and Taylor expanding $e^x$. Hence, there exists a $\hat{e}_i$, s.t. $\tilde{\lambda}_i^n = (1 + \hat{e}_i)\lambda_i^n$ and $|\hat{e}_i| \le 2n|e_i|$

$$
\begin{aligned}
\|\tilde{L}^{(n)}\tilde{f}_{\text{span}} - L^{(n)}f_{\text{span}}\|_{L^2(\Omega)} &= \left\|\sum_{i=1}^{k}\left(\tilde{\lambda}_i^n f_i\tilde{\varphi}_i - \lambda_i^n f_i\varphi_i\right)\right\|_{L^2(\Omega)} \\
&\le \left\|\sum_{i=1}^{k}\left((1 + \hat{e}_i)\lambda_i^n f_i\tilde{\varphi}_i - \lambda_i^n f_i\varphi_i\right)\right\|_{L^2(\Omega)} \\
&\le \left\|\sum_{i=1}^{k}(\lambda_i^n f_i\tilde{\varphi}_i - \lambda_i^n f_i\varphi_i)\right\|_{L^2(\Omega)} + \left\|\sum_{i=1}^{k}\hat{e}_i\lambda_i^n f_i\tilde{\varphi}_i\right\|_{L^2(\Omega)} \\
&\overset{(2)}{\le} \lambda_k^n\frac{\|f_{\text{span}}\|_{L^2(\Omega)}2^{3/2}\delta}{\gamma - \delta} + \lambda_k^n\max_i|\hat{e}_i|\|\tilde{f}_{\text{span}}\|_{L^2(\Omega)} \qquad (34)
\end{aligned}
$$

where we get (1) by Lemma 11 and (2) first using the fact that eigenvalues of $L$ are monotonically increasing, i.e., $\lambda_1 \le \lambda_2 \le \cdots \lambda_k$.

From (33) and (34) we have the following:

$$\|\tilde{L}^{(n)}(\tilde{f}_{\text{span}} - f_{\text{span}})\|_{L^2(\Omega)} \leq \lambda_k^n \frac{\|f_{\text{span}}\|_{L^2(\Omega)} 2^{3/2} \delta}{\gamma - \delta} + \lambda_k^n \max_i |\hat{e}_i| \|\tilde{f}_{\text{span}}\|_{L^2(\Omega)} + n 2 \delta \lambda_k^n \|f_{\text{span}}\|_{L^2(\Omega)}$$

Since $\delta \ll 1$ and also $e \ll 1$, we therefore have that

$$\|\tilde{L}^{(n)} \tilde{f}_{\text{span}} - L^{(n)} f_{\text{span}}\|_{L^2(\Omega)} \leq \lambda_k^n \frac{\|f_{\text{span}}\|_{L^2(\Omega)} 2^{3/2} \delta}{\gamma - \delta}$$

Combining with the result for Part 1, Therefore we have the following:

$$\|\tilde{L}^{(n)}(\tilde{f}_{\text{span}} - f_{\text{nn}})\|_{L^2(\Omega)} \leq (n!)^2 \cdot C^n (\epsilon_{\text{span}} + \epsilon_{\text{nn}}) + \lambda_k^n \frac{\|f_{\text{span}}\|_{L^2(\Omega)} 2^{3/2} \delta}{\gamma - \delta}$$

$\square$

# F   Technical Lemmas: Manipulating Operators

Before we state the lemmas we introduce some common notation used throughout this section. We denote $L^{(n)} = \underbrace{L \circ L \circ \cdots \circ L}_{n \text{ times}}$. Further we use $L_k$ to denote the operator with $\partial_k a_{ij}$ for all $i, j \in [d]$ and $\partial_k c$ as coefficients, that is:

$$L_k u = \sum_{i,j=1}^{d} -(\partial_k a_{ij}) \partial_{ij} u - \sum_{i,j=1}^{d} \partial_k (\partial_i a_i) \partial_j u + (\partial_k c) u$$

Similarly the operator $L_{kl}$ is defined as:

$$L_{kl} u = \sum_{i,j=1}^{d} -(\partial_{kl} a_{ij}) \partial_{ij} u - \sum_{i,j=1}^{d} \partial_{kl} (\partial_i a_i) \partial_j u + (\partial_{kl} c) u$$

**Lemma 13** (Relative operator perturbation bound). *Consider the operator $\tilde{L}$, defined in (4), then for all $u \in H_0^1(\Omega)$ the following holds,*

$$\langle (\tilde{L} - L) u, u \rangle \leq \delta \langle L u, u \rangle \tag{35}$$

*where $\delta = \frac{1}{\min\{m/\epsilon_A, \zeta/\epsilon_c\}}$.*

*Proof.*

$$\langle (\tilde{L} - L) u, u \rangle = \int_\Omega \left( (\tilde{A} - A) \nabla u \cdot \nabla u + (\tilde{c} - c) u^2 \right) dx$$

$$\leq \left( \max_{ij} \|\tilde{A}_{ij} - A_{ij}\|_{L^\infty(\Omega)} \right) \|\nabla u\|_{L^2(\Omega)}^2 + \|\tilde{c} - c\|_{L^\infty(\Omega)} \|u\|_{L^2(\Omega)}^2$$

$$\leq \epsilon_A \|\nabla u\|_{L^2(\Omega)}^2 + \epsilon_c \|u\|_{L^2(\Omega)}^2 \tag{36}$$

Further, note that

$$\langle L u, u \rangle = \int_\Omega A \nabla u \cdot \nabla u + c u^2 dx$$

$$\geq m \|\nabla u\|_{L^2(\Omega)}^2 + \zeta \|u\|_{L^2(\Omega)}^2 \tag{37}$$

Using the inequality $\frac{a+b}{c+d} \geq \min\{\frac{a}{c}, \frac{b}{d}\}$ from (36) and (37), we have

$$\frac{m \|\nabla u\|_{L^2(\Omega)}^2 + \zeta \|u\|_{L^2(\Omega)}^2}{\epsilon_A \|\nabla u\|_{L^2(\Omega)}^2 + \epsilon_c \|u\|_{L^2(\Omega)}^2} \geq \min\left\{ \frac{m}{\epsilon_A}, \frac{\zeta}{\epsilon_c} \right\} \tag{38}$$

Hence this implies that

$$\langle (\tilde{L} - L) u, u \rangle \leq \delta \langle L u, u \rangle$$

where $\delta = \frac{1}{\min\{m/\epsilon_A, \zeta/\epsilon_c\}}$. $\square$

**Lemma 14** (Operator Chain Rule). *Given an elliptic operator L, for all $v \in C^\infty(\Omega)$ we have the following*

$$\nabla_k L^{(n)} u = \sum_{i=1}^{n} \left( L^{(n-i)} \circ L_k \circ L^{(i-1)} \right)(u) + L^{(n)}(\nabla_k u) \tag{39}$$

$$
\begin{aligned}
\nabla_{kl}(L^{(n)}u) = &\sum_{\substack{i,j \\ i<j}} \left( L^{n-i} \circ L_k \circ L^{(j-i-1)} \circ L_l \circ L^{j-1} \right) u \\
&+ \sum_{\substack{i,j \\ i>j}} \left( L^{n-j} \circ L_k \circ L^{(i-j-1)} \circ L_l \circ L^{i-1} \right) u \\
&+ \sum_{i} \left( L^{n-i} \circ L_{kl} \circ L^{i-1} \right) u + L^{(n)}(\nabla_{kl}u)
\end{aligned}
\tag{40}
$$

*where we assume that $L^{(0)} = I$.*

*Proof.* We show the proof using induction on $n$. To handle the base case, for $n = 1$, we have

$$
\begin{aligned}
\nabla_k (Lu) &= \nabla_k \left( -\mathrm{div}(A\nabla u) + cu \right) \\
&= \nabla_k \left( -\sum_{ij} a_{ij}\partial_{ij}u - \sum_{ij} \partial_i a_{ij}\partial_j u + cu \right) \\
&= \left( -\sum_{ij} a_{ij}\partial_{ij}(\partial_k u) - \sum_{ij} \partial_i a_{ij}\partial_j \partial_k u + c\partial_k u \right) \\
&\quad + \left( -\sum_{ij} \partial_k a_{ij}\partial_{ij}u - \sum_{ij} \partial_i \partial_k a_{ij}\partial_j u + \partial_k cu \right) \\
&= L(\nabla_k u) + L_k u
\end{aligned}
\tag{41}
$$

Similarly $n = 1$ and $k, l \in [d]$,

$$
\begin{aligned}
\nabla_{kl}(Lu) &= \nabla_{kl} \left( -\mathrm{div}(A\nabla u) + cu \right) \\
&= \nabla_{kl} \left( -\sum_{ij} a_{ij}\partial_{ij}u - \sum_{ij} \partial_i a_{ij}\partial_j u + cu \right) \\
&= \left( -\sum_{ij} a_{ij}\partial_{ij}(\partial_{kl}u) - \sum_{ij} \partial_i a_{ij}\partial_j \partial_{kl}u + c\partial_{kl}u \right) \\
&\quad + \left( -\sum_{ij} \partial_k a_{ij}\partial_{ij}\partial_l u - \sum_{ij} \partial_i \partial_k a_{ij}\partial_j \partial_l u + \partial_k c\partial_l u \right) \\
&\quad + \left( -\sum_{ij} \partial_l a_{ij}\partial_{ij}\partial_k u - \sum_{ij} \partial_i \partial_l a_{ij}\partial_j \partial_k u + \partial_l c\partial_k u \right) \\
&\quad + \left( -\sum_{ij} \partial_{kl} a_{ij}\partial_{ij}u - \sum_{ij} \partial_i \partial_{kl} a_{ij}\partial_j u + \partial_{kl} cu \right) \\
&= L(\nabla_{kl}u) + L_k(\nabla_l u) + L_l(\nabla_k u) + L_{kl} u
\end{aligned}
\tag{42}
$$

For the inductive case, assume that for all $m < n$, (39) and (40) hold. Then, for any $k \in [d]$ we have:

$$\nabla_k(L^{(n)}u) = \nabla_k \left( L \circ L^{(n-1)}(u) \right)$$

$$= L \left( \nabla_k(L^{(n-1)}u) \right) + L_k \left( L^{(n-1)}u \right)$$

$$= L \left( \sum_{i=1}^{n-1} \left( L^{(n-1-i)} \circ L_k \circ L^{(i-1)} \right) u + L^{(n-1)}(\nabla_k u) \right) + L_k \left( L^{(n-1)} \right) u$$

$$= \sum_{i=1}^{n} \left( L^{(n-i)} \circ L_k \circ L^{(i-1)} \right)(u) + L^{(n)}(\nabla_k u) \tag{43}$$

Similarly, for all $k, l \in [d]$ we have:

$$\nabla_{kl}(L^{(n)}u) = \nabla_{kl} \left( L \circ L^{(n-1)}(u) \right)$$

$$= L \left( \nabla_{kl}(L^{(n-1)}u) \right) + L_k \left( \nabla_l \left( L^{(n-1)}u \right) \right) + L_l \left( \nabla_k \left( L^{(n-1)}u \right) \right) + L_{kl} \left( L^{(n-1)}u \right)$$

$$= L \Bigg( \sum_{\substack{i,j \\ i<j}}^{n-1} \left( L^{(n-1-i)} \circ L_k \circ L^{(j-i-1)} \circ L_l \circ L^{(j-1)} \right) u$$

$$+ \sum_{\substack{i,j \\ i>j}}^{n-1} \left( L^{(n-1-j)} \circ L_k \circ L^{(i-j-1)} \circ L_l \circ L^{(i-1)} \right) u$$

$$+ \sum_{i=1}^{n-1} \left( L^{(n-1-i)} \circ L_{kl} \circ L^{(i-1)} \right) u + L^{(n-1)}(\nabla_{kl} u) \Bigg)$$

$$+ L_k \left( \sum_{i=1}^{n-1} \left( L^{(n-1-i)} \circ L_l \circ L^{(i-1)} \right)(u) + L^{(n-1)}(\nabla_l u) \right) \qquad \text{(from (43))}$$

$$+ L_l \left( \sum_{i=1}^{n-1} \left( L^{(n-1-i)} \circ L_k \circ L^{(i-1)} \right)(u) + L^{(n-1)}(\nabla_k u) \right) \qquad \text{(from (43))}$$

$$+ L_{kl} \left( L^{(n-1)}u \right)$$

$$= \sum_{\substack{i,j \\ i<j}}^{n} \left( L^{(n-i)} \circ L_k \circ L^{(j-i-1)} \circ L_l \circ L^{j-1} \right) u$$

$$+ \sum_{\substack{i,j \\ i>j}}^{n} \left( L^{(n-j)} \circ L_k \circ L^{(i-j-1)} \circ L_l \circ L^{(i-1)} \right) u$$

$$+ \sum_{i}^{n} \left( L^{(n-i)} \circ L_{kl} \circ L^{(i-1)} \right) u + L^{(n)}(\nabla_{kl} u) \tag{44}$$

By induction, the claim follows. $\qquad\qquad\qquad\qquad\qquad\qquad\qquad\qquad\qquad\qquad \square$

**Lemma 15.** *For all $u \in C^{\infty}(\Omega)$ then for all $k, l \in [d]$ the following upper bounds hold,*

$$\|Lu\|_{L^2(\Omega)} \leq C \max_{\alpha:|\alpha|\leq 2} \|\partial^\alpha u\|_{L^2(\Omega)} \tag{45}$$

$$\|\nabla_k(Lu)\|_{L^2(\Omega)} \leq 2 \cdot C \max_{\alpha:|\alpha|\leq 3} \|\partial^\alpha u\|_{L^2(\Omega)} \tag{46}$$

*and*

$$\|\nabla_{kl}(Lu)\|_{L^2(\Omega)} \leq 4 \cdot C \max_{\alpha:|\alpha|\leq 4} \|\partial^\alpha u\|_{L^2(\Omega)} \tag{47}$$

*where*

$$C := (2d^2 + 1) \max \left\{ \max_{\alpha:|\alpha|\leq 3} \max_{i,j} \|\partial^\alpha a_{ij}\|_{L^\infty(\Omega)}, \max_{\alpha:|\alpha|\leq 2} \|\partial^\alpha c\|_{L^\infty(\Omega)} \right\}.$$

*Proof.* We first show the upper bound on $\|Lu\|_{L^2(\Omega)}$:

$$\|Lu\|_{L^2(\Omega)} \leq \left\| -\sum_{i,j=1}^{d} a_{ij}\partial_{ij}u - \sum_{i,j=1}^{d} \partial_i a_{ij}\partial_j u + cu \right\|_{L^2(\Omega)}$$

$$\leq^{(1)} \underbrace{(2d^2+1)\max\left\{\max_{i,j}\|\partial_i a_{ij}\|_{L^\infty(\Omega)}, \max_{i,j}\|a_{ij}\|_{L^\infty(\Omega)}, \|c\|_{L^\infty(\Omega)}\right\}}_{C_1}\max_{\alpha:|\alpha|\leq 2}\|\partial^\alpha u\|_{L^2(\Omega)}$$

$$\leq C_1 \max_{\alpha:|\alpha|\leq 2}\|\partial^\alpha u\|_{L^2(\Omega)} \tag{48}$$

where (1) follows by Hölder.

Proceeding to $\|\nabla_k(Lu)\|_{L^2(\Omega)}$, from Lemma16 we have

$$\|\nabla_k(Lu)\|_{L^2(\Omega)} \leq \|L_k u\|_{L^2(\Omega)} + \|L(\nabla_k u)\|_{L^2(\Omega)}$$

$$\leq \left\| -\sum_{i,j=1}^{d} \partial_k a_{ij}\partial_{ij}u - \sum_{i,j=1}^{d} \partial_{ik}a_{ij}\partial_j u + \partial_k cu \right\|_{L^2(\Omega)}$$

$$+ \left\| -\sum_{i,j=1}^{d} a_{ij}\partial_{ijk}u - \sum_{i,j=1}^{d} \partial_i a_{ij}\partial_{jk} u + c\partial_k u \right\|_{L^2(\Omega)}$$

$$\leq (2d^2+1)\max\left\{\max_{\alpha:|\alpha|\leq 2}\max_{i,j}\|\partial^\alpha a_{ij}\|_{L^\infty(\Omega)}, \|\partial_k c\|_{L^\infty(\Omega)}\right\}\max_{\alpha:|\alpha|\leq 2}\|\partial^\alpha u\|_{L^2(\Omega)}$$

$$+ (2d^2+1)\max\left\{\max_{\alpha:|\alpha|\leq 1}\max_{i,j}\|\partial^\alpha a_{ij}\|_{L^\infty(\Omega)}, \|c\|_{L^\infty(\Omega)}\right\}\max_{\alpha:|\alpha|\leq 3}\|\partial^\alpha u\|_{L^2(\Omega)}$$

$$\implies \|\nabla_k(Lu)\|_{L^2(\Omega)} \leq 2\cdot\underbrace{(2d^2+1)\max\left\{\max_{\alpha:|\alpha|\leq 2}\max_{i,j}\|\partial^\alpha a_{ij}\|_{L^\infty(\Omega)}, \max_{\alpha:|\alpha|\leq 1}\|\partial^\alpha c\|_{L^\infty(\Omega)}\right\}}_{C_2}\max_{\alpha:|\alpha|\leq 3}\|\partial^\alpha u\|_{L^2(\Omega)}$$

$$\leq 2\cdot C_2 \max_{\alpha:|\alpha|\leq 3}\|\partial^\alpha u\|_{L^2(\Omega)} \tag{49}$$

We use the result from Lemma 14 (equation (42)), to upper bound the quantity $\|\nabla_{kl}(Lu)\|_{L^2(\Omega)}$

$$\|\nabla_{kl}(Lu)\|_{L^2(\Omega)} \leq \|L_{kl}u\|_{L^2(\Omega)} + \|L_k(\nabla_l u)\|_{L^2(\Omega)} + \|L_l(\nabla_k u)\|_{L^2(\Omega)} + \|L(\nabla_{kl}u)\|_{L^2(\Omega)}$$

$$\leq \left\| -\sum_{i,j=1}^{d} \partial_{kl}a_{ij}\partial_{ij}u - \sum_{i,j=1}^{d} \partial_{ikl}a_{ij}\partial_j u + \partial_{kl}cu \right\|_{L^2(\Omega)}$$

$$+ \left\| -\sum_{i,j=1}^{d} \partial_k a_{ij}\partial_{ij}\partial_l u - \sum_{i,j=1}^{d} \partial_i\partial_k a_{ij}\partial_j \partial_l u + \partial_k c\partial_l u \right\|_{L^2(\Omega)}$$

$$+ \left\| -\sum_{i,j=1}^{d} \partial_l a_{ij}\partial_{ij}\partial_k u - \sum_{i,j=1}^{d} \partial_i\partial_l a_{ij}\partial_j \partial_k u + \partial_l c\partial_k u \right\|_{L^2(\Omega)}$$

$$+ \left\| -\sum_{i,j=1}^{d} a_{ij}\partial_{ijkl}u - \sum_{i,j=1}^{d} \partial_i a_{ij}\partial_{jkl} u + c\partial_{kl} u \right\|_{L^2(\Omega)}$$

$$\leq (2d^2+1)\max\left\{\max_{\alpha:|\alpha|\leq 3}\max_{i,j}\|\partial^\alpha a_{ij}\|_{L^\infty(\Omega)}, \|\partial_{kl}c\|_{L^\infty(\Omega)}\right\}\max_{\alpha:|\alpha|\leq 2}\|\partial^\alpha u\|_{L^2(\Omega)}$$

$$+ 2(2d^2+1)\max\left\{\max_{\alpha:|\alpha|\leq 2}\max_{i,j}\|\partial^\alpha a_{ij}\|_{L^\infty(\Omega)}, \|c\|_{L^\infty(\Omega)}\right\}\max_{\alpha:|\alpha|\leq 3}\|\partial^\alpha u\|_{L^2(\Omega)}$$

$$+ (2d^2+1)\max\left\{\max_{\alpha:|\alpha|\leq 2}\max_{i,j}\|\partial^\alpha a_{ij}\|_{L^\infty(\Omega)}, \|c\|_{L^\infty(\Omega)}\right\}\max_{\alpha:|\alpha|\leq 4}\|\partial^\alpha u\|_{L^2(\Omega)}$$

$$\implies \|\nabla_{kl}(Lu)\|_{L^2(\Omega)} \leq 4\cdot\underbrace{(2d^2+1)\max\left\{\max_{\alpha:|\alpha|\leq 3}\max_{i,j}\|\partial^\alpha a_{ij}\|_{L^\infty(\Omega)}, \max_{\alpha:|\alpha|\leq 2}\|\partial^\alpha c\|_{L^\infty(\Omega)}\right\}}_{C_3}\max_{\alpha:|\alpha|\leq 4}\|\partial^\alpha u\|_{L^2(\Omega)}$$

$$\leq 4\cdot C_3 \max_{\alpha:|\alpha|\leq 4}\|\partial^\alpha u\|_{L^2(\Omega)} \tag{50}$$

Since $C_1 \leq C_2 \leq C_3$, we define $C := C_3$ and therefore from equations (48), (49) and (50) the claim follows.

Further, we note that from (49), we also have that

$$\|L_k(u)\|_{L^2(\Omega)}, \|L(\nabla_k u)\|_{L^2(\Omega)} \leq C \max_{\alpha:|\alpha|\leq 3} \|\partial^\alpha u\|_{L^2(\Omega)} \tag{51}$$

and similarly from (50) we have that,

$$\|L_{kl}(u)\|_{L^2(\Omega)}, \|L_k(\nabla_l u)\|_{L^2(\Omega)}, \|L_l(\nabla_k u)\|_{L^2(\Omega)}, \|L(\nabla_{kl} u)\|_{L^2(\Omega)} \leq C \max_{\alpha:|\alpha|\leq 4} \|\partial^\alpha u\|_{L^2(\Omega)} \tag{52}$$

$\square$

**Lemma 16.** *For all $u \in C^\infty(\Omega)$ and $k, l \in [d]$ then for all $n \in \mathbb{N}$ we have the following upper bounds,*

$$\|L^{(n)} u\|_{L^2(\Omega)} \leq (n!)^2 \cdot C^n \max_{\alpha:|\alpha|\leq n+2} \|\partial^\alpha u\|_{L^2(\Omega)} \tag{53}$$

$$\|\nabla_k(L^{(n)} u)\|_{L^2(\Omega)} \leq (n+1) \cdot (n!)^2 \cdot C^n \max_{\alpha:|\alpha|\leq n+2} \|\partial^\alpha u\|_{L^2(\Omega)} \tag{54}$$

$$\|\nabla_{kl}(L^{(n)} u)\|_{L^2(\Omega)} \leq ((n+1)!)^2 \cdot C^n \max_{\alpha:|\alpha|\leq n+3} \|\partial^\alpha u\|_{L^2(\Omega)} \tag{55}$$

*where $C = (2d^2 + 1) \max \left\{ \max_{\alpha:|\alpha|\leq 3} \max_{i,j} \|\partial^\alpha a_{ij}\|_{L^\infty(\Omega)}, \max_{\alpha:|\alpha|\leq 2} \|\partial^\alpha c\|_{L^\infty(\Omega)} \right\}.$*

*Proof.* We prove the Lemma by induction on $n$. The base case $n = 1$ follows from Lemma 15, along with the fact that $\max_{\alpha:|\alpha|\leq 2} \|\partial^\alpha u\|_{L^2(\Omega)} \leq \max_{\alpha:|\alpha|\leq 3} \|\partial^\alpha u\|_{L^2(\Omega)}$.

To show the inductive case, assume that the claim holds for all $m \leq (n-1)$. By Lemma 15, we have

$$\|L^{(n)} u\|_{L^2(\Omega)} = \|L(L^{(n-1)} u)\|_{L^2(\Omega)}$$

$$\leq \left\| -\sum_{i,j=1}^d a_{ij} \partial_{ij}(L^{(n-1)} u) - \sum_{i,j=1}^d \partial_i a_{ij} \partial_j(L^{(n-1)} u) + c(L^{(n-1)} u) \right\|_{L^2(\Omega)}$$

$$\leq C \cdot \max \left\{ \|L^{(n-1)} u\|_{L^2(\Omega)}, \max_i \|\nabla_i(L^{(n-1)} u)\|_{L^2(\Omega)}, \max_{i,j} \|\nabla_{ij}(L^{(n-1)} u)\|_{L^2(\Omega)} \right\}$$

$$\leq C \cdot (n!)^2 \cdot C^{n-1} \max_{\alpha:|\alpha|\leq (n-1)+3} \|\partial^\alpha u\|_{L^2(\Omega)}$$

Thus, we have

$$\|L^{(n)} u\|_{L^2(\Omega)} \leq (n!)^2 \cdot C^n \max_{\alpha:|\alpha|\leq n+2} \|\partial^\alpha u\|_{L^2(\Omega)}$$

as we need.

Similarly, for $k \in [d]$, we have:

$$\|\nabla_k(L^{(n)} u)\|_{L^2(\Omega)} \leq \sum_{i=1}^n \left\| \left( L^{(n-i)} \circ L_k \circ L^{(i-1)} \right)(u) \right\|_{L^2(\Omega)} + \|L^{(n)}(\nabla_k u)\|_{L^2(\Omega)}$$

$$\leq (n) \cdot (n!)^2 \cdot C^n \max_{\alpha:|\alpha|\leq n+2} \|\partial^\alpha u\|_{L^2(\Omega)} + (n!)^2 \cdot C^n \max_{\alpha:|\alpha|\leq n+2} \|\partial^\alpha u\|_{L^2(\Omega)}$$

$$\leq (n+1) \cdot (n!)^2 \cdot C^n \max_{\alpha:|\alpha|\leq n+2} \|\partial^\alpha u\|_{L^2(\Omega)} \tag{56}$$

Finally, for $k, l \in [d]$ we have

$$
\begin{aligned}
\|\nabla_{kl}(L^{(n)}u)\|_{L^2(\Omega)} &\leq \sum_{\substack{i,j \\ i<j}} \left\|\left(L^{(n-i)} \circ L_k \circ L^{(j-i-1)} \circ L_l \circ L^{(j-1)}\right) u\right\|_{L^2(\Omega)} \\
&\quad + \sum_{\substack{i,j \\ i>j}} \left\|\left(L^{(n-j)} \circ L_k \circ L^{(i-j-1)} \circ L_l \circ L^{(i-1)}\right) u\right\|_{L^2(\Omega)} \\
&\quad + \sum_{i} \left\|\left(L^{(n-i)} \circ L_{kl} \circ L^{(i-1)}\right) u\right\|_{L^2(\Omega)} + \|L^{(n)}(\nabla_{kl}u)\|_{L^2(\Omega)} \\
&\leq n(n+1) \cdot (n!)^2 \cdot C^n \max_{\alpha:|\alpha|\leq n+2} \|\partial^\alpha u\|_{L^2(\Omega)} \\
&\quad + n \cdot (n!)^2 \cdot C^n \max_{\alpha:|\alpha|\leq n+2} \|\partial^\alpha u\|_{L^2(\Omega)} + C^n \max_{\alpha:|\alpha|\leq n+3} \|\partial^\alpha u\|_{L^2(\Omega)} \\
\implies \|\nabla_{kl}(L^{(n)}u)\|_{L^2(\Omega)} &\leq ((n+1)!)^2 \cdot C^n \max_{\alpha:|\alpha|\leq n+3} \|\partial^\alpha u\|_{L^2(\Omega)}
\end{aligned}
$$

$$(57)$$

Thus, the claim follows. $\qquad \square$

**Lemma 17.** *Let $A_n^{(i)}$, $i \in [n]$ be defined as a composition of $(n - i)$ applications of $L$ and $i$ applications of $L \circ \Sigma$ (in any order), s.t. $\|\Sigma\| \leq \delta$. Then, we have:*

$$
\|L^{-(n)} A_n^{(i)}\| \leq \delta^i \qquad (58)
$$

*Proof.* We prove the above claim by induction on $n$.

For $n = 1$ we have two cases. If $A^{(1)} = L \circ \Sigma$, we have:

$$
\|L^{-1} \circ L \circ \Sigma\| \leq \delta
$$

If $A^{(1)} = L$ we have:

$$
\|L^{-1} L\| = 1
$$

Towards the inductive hypothesis, assume that for $m \leq n - 1$ and $i \in [n - 1]$ it holds that,

$$
\|L^{(n-1)} A_{(n-1)}^{(i)}\| \leq \delta^i
$$

For $n$, we will have two cases. First, if $A_{(n)}^{(i+1)} = A_{(n-1)}^{(i)} \circ L \circ \Sigma$, by submultiplicativity of the operator norm, as well as the fact that similar operators have identical spectra (hence equal operator norm) we have:

$$
\begin{aligned}
\|L^{-(n)} \circ A_n^{(i+1)}\| &= \|L^{-1} \circ L^{-(n-1)} \circ A_{n-1}^{(i)} \circ L \circ \Sigma\| \\
&= \|L^{-(n-1)} \circ A_{n-1}^{(i)} \circ L \circ \Sigma \circ L^{-1}\| \\
&\leq \delta \|L^{-(n-1)} A_{(n-1)}^{(i-1)}\| \|L \circ \Sigma \circ L^{-1}\| \\
&\leq \delta^i \delta = \delta^{i+1}
\end{aligned}
$$

so the inductive claim is proved. In the second case, $A_{(n)}^{(i)} = A_{(n-1)}^{(i)} L$ and we have, by using the fact that the similar operators have identical spectra:

$$
\begin{aligned}
\|L^{(-n)} \circ A_{(n)}^{(i)} \circ L\| &= \|L^{-(n-1)} \circ A_{(n-1)}^{(i)} \circ L \circ L^{-1}\| \\
&= \|L^{-(n-1)} \circ A_{(n-1)}^{(i)}\| \leq \delta^i
\end{aligned}
$$

where the last inequality follows by the inductive hypothesis. $\qquad \square$