# OpenReview forum: "Parametric Complexity Bounds for Approximating PDEs with Neural Networks"
_NeurIPS.cc/2021/Conference — NeurIPS 2021 Spotlight_

### Official Review · Reviewer_WhbH · 2021-07-05

**Rating:** 6
**Confidence:** 4

**Summary:**

The authors attempt to bound a neural network's complexity that approximates a solution to a self-adjoint elliptic PDE. The paper lacks mathematical rigor on the PDE theory side: the most important one is that their gradient descent step doesn't make sense from a functional analysis point-of-view (see main review). The authors need to compare the complexity of their neural network to the bound obtained by combining standard PDE regularity theory (see Evan's Section 6.3) with the complexity of neural networks for approximating smooth functions. Since the main theorem doesn't take into account the smoothness of the boundary of \Omega, I suspect that the latter gives much stronger bounds for many practical domains.

**Limitations And Societal Impact:**

Some limitations are discussed.

**Main Review:**

Definition 1 is for self-adjoint elliptic PDEs, not general elliptic PDEs.

The domain \Omega is assumed to be a bounded open set with a boundary. However, to define H_0^1(\Omega), the domain must have a bounded trace operator. The usual assumption is that the domain has a Lipschitz boundary.

The smoothness of the boundary of the domain *greatly* affects the smoothness of the PDE solution. The paper gives the impression that "if an elliptic PDE's coefficients and forcing term are easy to approximate by a neural network, then so is the PDE's solution." However, this is forgetting about the important role of the smoothness of the domain's boundary. Consider Poisson's equation on the unit square [0,1]^2 with zero Dirichlet conditions with forcing term f = 1. The coefficients of Poisson's equation and forcing term are trivial to represent by a neural network. However, the solution has logarithmic singularities at the four corners of the domain. In contrast, consider Poisson's equation on the unit disk with zero Dirichlet conditions with forcing term f = 1. Then, the solution is polynomial: u(x,y) = (x^2+y^2 - 1)/4. The main theorem ignores the smoothness of the boundary of \Omega and without that I suspect that one can do better by combining standard PDE regularity theory (see Evan's Section 6.3) with the complexity of neural networks for approximating smooth functions.

Unfortunately, the gradient descent step in (5) seems to assume that L can be applied to functions in strong form. In strong form, the elliptic PDE operator L takes derivatives so it reduces the regularity of the gradient descent iterates. One cannot take two steps of the gradient descent iteration, as the second iterate may not be twice weakly differentiable. For the gradient step to make sense, the authors need to assume that the initial guess is infinity differentiable.


**Time Spent Reviewing:**

3

---

> ### Author Response · Authors · 2021-08-10
> **Reply to Reviewer WhbH**
>
> We thank the reviewer for their detailed feedback and comments. In the revised version, we will clearly state that Definition 1 pertains to self-adjoint elliptic PDEs and not general elliptic PDEs.
>
>
> **Re: The main theorem ignores the smoothness of the boundary of \Omega**\
> While it might seem intuitive that *sufficiently smooth* functions could be approximated by neural networks with a “small” size, this is not generally true. Namely, smooth functions can be approximated over a compact domain [Hornik et al, 1989], but in general, the size of the network will grow exponentially in the dimensions (i.e. will incur a “curse of dimensionality”). Thus, bootstrapping onto classical results characterizing the regularity of the solutions of PDEs (e.g., Evans) is of no help (there is no way to relate the size of the final network to the size of the coefficient networks).
>
> Coincidentally, some recent approximation results [Lu et al 2021] use this intuition: they show that if the coefficients of the PDE have a small “Barron norm”, the solution also has a small “Barron norm”—and the Barron norm of a function can be used to bound the size of a shallow (2-layer) network needed to approximate it. Importantly though: purely analytic results (i.e., merely leveraging a handle on the regularity of the function) seem fundamentally unable to make use of the neural network complexity of the coefficients, which motivated us to explore new avenues for proof techniques.
>
> We agree that *implicitly*, the “niceness” of the domain matters—it would come in through the fact that by Assumptions 1 and 3, $f$ is close to $f_\mbox{span}$ (which is a linear combination of the top $k$ Dirichlet eigenfunctions of $L$, which heavily depend on the domain) and $f$ is approximable by a small neural network—hence, by triangle inequality  $f_\mbox{span}$ is approximable by a small neural network.
>
> **Re: need to assume that the initial guess is infinitely differentiable**\
> Yes, good catch! We will add explicitly the requirement that $u_0$ has activation functions in  $\Sigma \cup \Sigma’ \cup \{\rho\}$ (which would imply that it is infinitely differentiable).
>
> ---
> [1] Hornik, Kurt, Maxwell Stinchcombe, and Halbert White. "Multilayer feedforward networks are universal approximators." Neural networks 2.5 (1989): 359-366.
>
> [2] Lu, Jianfeng, Yulong Lu, and Min Wang. "A priori generalization analysis of the deep ritz method for solving high dimensional elliptic equations." arXiv preprint arXiv:2101.01708 (2021).

---

> ### Author Response · Authors · 2021-08-22
> **Can we help address some other concerns?**
>
> We thank you again for the thoughtful review. Did our reply help address your concerns? Please let us know if there are some other questions we can resolve.

---

> ### Comment · Area_Chair_6aXs · 2021-08-26
> **Please respond to the correspondence of the authors**
>
> Dear Reviewer,
>
> It would be great to have your response to authors' rebuttal remarks.
>
> In particular, they argue that they can assume the infinite differentiability condition you mentioned and also provide extenuating conditions on the domain issue you brought up.
>
> best,
> AC

---

> ### Comment · Reviewer_WhbH · 2021-08-29
> **Reply**
>
> Yes, the authors have answered many of the concerns about correctness.
>
> Still, I find it difficult to assess if the derived bounds are useful. I don't understand how the author's comments about f and f_span resolves my concerns about the smoothness of the domain and usefulness of the bounds. The eigenfunctions of the Laplacian on the disk with zero Dirichlet conditions are related to Bessel functions and Bessel series converge very close to many smooth functions. This is why no efficient numerical solvers for PDEs on the disk are built from Bessel functions and spectral methods don't always use eigenfunctions of the operator. I guess this means that the gap between f and f_span would be large for moderate k.

---

> > ### Author Response · Authors · 2021-08-30
> > **Re: Reply**
> >
> > Thank you for your reply. We’re glad to hear that our first reply resolved many of your concerns about correctness and hope that the following response can resolve your concerns about the usefulness of our results.
> >
> > Re: smoothness, we would like to further clarify the following two points from our previous response:
> >
> > 1. *Smoothness itself (whether on the domain or the coefficients) is not enough to avoid the curse of dimensionality in terms of neural network approximability.* The original review stated “The authors need to compare the complexity of their neural network to the bound obtained by combining standard PDE regularity theory (see Evan's Section 6.3) with the complexity of neural networks for approximating smooth functions.” Our point in the response was that, in general, approximating smooth functions (even $C^{\infty}$ functions, with bounded derivatives in $\Omega$) requires exponentially sized (in dimension) neural networks. Thus, the regularity of the solution alone is insufficient.
> > Our results are useful because they provide a set of conditions under which approximating the solutions (provably) exhibit a less than exponential dependence on the dimension of the domain.
> > 2.  *The domain implicitly enters into our results (i.e. for some domains, our results will yield a smaller network vs for others)*  This is because we require f (and hence, f_span) to be approximable by a neural network and f_span depends on the Dirichlet eigenfunctions, which depend upon the domain. We hope that this addresses the question that our result ignores the relevance of the boundary.
> >
> > Concerning your example where the domain is the unit disk, it seems that you are asking *when in practice would you use a spectral method (with a potentially “user-specified” basis)* versus a neural network. This is a question of inductive bias: if the PDE in question is such that, in some nice basis, the coefficients of the PDE are approximable by a *small number* of basis elements in a computationally convenient basis, using a spectral method could be appropriate. Our result characterizes when *neural networks* have a good inductive bias—namely, when the coefficients of the PDE are approximable by small neural networks. This is analogous, in the classical supervised learning setting, to asking why one would use neural networks versus judiciously engineered kernels (e.g. in vision, using hand-crafted embeddings like SIFT/HOG): depending on the structure of the *specific task*, it may well be that kernels do equally as well or outperform neural networks.

---

> ### Comment · Reviewer_WhbH · 2021-08-30
> **Rating increased**
>
> Since my correctness concerns have been fully resolved, I have increased my rating.  I think the revisions pointed out by the other reviewers will also improve the clarity of the manuscript.  I would like to thank the authors for their careful and thoughtful replies.

---

### Official Review · Reviewer_rCSp · 2021-07-10

**Rating:** 7
**Confidence:** 4

**Summary:**

This paper aims to establish neural network representation results for solutions to elliptic PDEs. The main result states that such solutions can be approximated without curse of dimensionality.

**Ethical Concerns:**

Everything ok in this respect

**Limitations And Societal Impact:**

Everything ok in this respect

**Main Review:**

The approximation of high dimensional PDEs is a very relevant topic and neural networks are undoubtedly a very promising tool for this problem. There is already a significant body of research confirming that neural networks do not suffer from the curse of dimensionality. However, the approach of these previous works is different from the auhor's approach. In the present paper

1.  the solution of an elliptic PDE is formulated as stationary limit of a parabolic flow that can be discretized yielding a sequence of functions, obtained form iteratively applying a suitable differential operator, that converges exponentially to the PDE solution.
2. it is "shown" that applying one iteration step does not have a tangible negative impact on the approximability of a function by neural networks.
3. together with the exponential convergence of the iteration, the approximation result follows.



Points 1. and 3. are certainly well known. Unfortunately, 2. is not correct. The authors state a Lemma (Lemma 7 in their paper) that claims that the derivative of a neural network is also a neural network of comparable size. This is wrong. The hat function is a small ReLU network, its derivative is discontinuous and therefore cannot be represented as a ReLU network. The second derivative is a linear combination of Dirac distributions - not even a function anymore.


------
After discussing with the authors I am now confident in the correctness of the claimed result. The issue has been that the authors use a highly unusual definition of neural networks which allows to choose at every node a different activation function from a countable set of activation functions. This set of activation functions consists of all derivatives of a given C^\infty function, as well as the square function. This casts some doubt regarding the relevance of this result because it is clearly impossible to train such networks. The result is thus purely theoretical, even much more than "usual" neural network approximation results.

Overall, given these considerations I am raising my score to 5.

------
After more discussions, the authors could also address the concern related to the unusual choice of activation function.

I thank the authors for putting so such effort into this clarifying process and raise my score to 7.

**Time Spent Reviewing:**

7

---

> ### Author Response · Authors · 2021-08-10
> **Reply to Reviewer rCSp**
>
> We thank the reviewer for their comments.
>
> We would like to point out that Assumption (iii) (lines 159-160) restricts the activations used to be infinitely differentiable.  Since the ReLU activation is not infinitely differentiable, the scenario that presented in the review is **not a counterexample** to our theorem.
>
> At the same time, we agree that these details should be stated clearly in Lemma 7. We will update the final version of our paper to say that if g has differentiable activation functions $\{\sigma_i\}$, then the neural network for the derivative uses activation functions $\{\sigma_i, \sigma’_i\}$. (In fact, this is why in our theorem statement the final neural network uses $\Sigma$ and $\Sigma’$ as the activation functions.)
>
> Finally, as we discuss at length in the general reply to all reviewers, the choice of activations is somewhat immaterial to the final results and can be chosen for mathematical convenience.

---

> > ### Comment · Reviewer_rCSp · 2021-08-10
> > **Differentiation of Neural Networks**
> >
> > even if the activation function is infinitely differentiable:
> >
> > 1. the derivative will be a neural network with a *different* activation function
> > 2. the size of the neural network will grow
> >
> > during the evolution, both 1. and 2. occur at every step which invalidates the bounds on the network size and the activation function changes repeatedly! what exactly would then be the activation function of the final network?
> >
> > summarizing, i am still *strongly convinced* that the proof of the stated result is false!

---

> > > ### Author Response · Authors · 2021-08-10
> > > **Re:Differentiation of Neural Networks**
> > >
> > > Thank you for your response.
> > >
> > > * The question asked is in fact **answered in the paper**: the final network has activation functions in the set $\Sigma \cup \Sigma’ \cup \{\rho\}$, where $\Sigma’$ is the set of *all higher-order derivatives* of the activations in $\Sigma$ (lines 159-161). This is exactly because, as you noted, derivatives are being applied repeatedly at each gradient descent step.
> > >
> > > * It is **true** that the network grows at each gradient descent step—that is how the proof proceeds! Theorem 1 is proven by (i) **bounding the size increase at every gradient step** (Lemma 4); (ii) bounding the number of gradient steps needed (Lemma 3).

---

> > > > ### Comment · Reviewer_rCSp · 2021-08-11
> > > > **Re: Re: Differentiation of Neural Networks**
> > > >
> > > > Thank you for this clarification.
> > > >
> > > > Thank you also for pointing out Lemma 4 which indeed addresses the growth of the neural network size during the iteration **if the validity of Lemma 7 is assumed**.
> > > >
> > > > Lemma 7 is however formulated without any mathematical rigor (there is not even a formal definition of a neural network) and as reference for a proof the authors cite the original 4 page backprop paper which does not contain any mathematical statement. Note that the claimed result is not trivial, especially if the precise growth of the network size needs to be controlled. It is also not so clear to me that the derivative network is still a classical neural network (computing the derivative requires taking products of different sub networks...).
> > > >
> > > > Since this is actually *the* crucial result of the paper (the other results are well known) which is stated without a precise formulation and without proof, it is difficult to accept the presentation in the paper as a proof.

---

> > > > > ### Author Response · Authors · 2021-08-12
> > > > > **Re: Lemma 7**
> > > > >
> > > > > Thank you for your quick response. We are glad to know that we have narrowed down the source of worry to Lemma 7. First off, we agree that Lemma 7 ought to be self-contained and include details regarding the activation functions of the network, and below we present an updated version that we hope will resolve this concern. Second, we would like to stress that Lemma 7 is **not a core contribution** of our paper—it is well-known and used in many other (recent works) in almost the same form we are using: e.g., [Bai et al (2018), Lemma E.7; Agarwal et al (2017), Claim A.1]. We cited Rumelhart because it is the standard reference most familiar to the machine learning community (the above two works cite the same paper as well).
> > > > >
> > > > > We now present our proposed **modification to the statement of Lemma 7** as well as a **self-contained proof of Lemma 7** that we will add to the paper for completeness. The proof is almost exactly the same as Chen et al (2010)’s proof (of their **Theorem 9.10**). And that proof, in turn, adapts the proof of the famous Baur-Strassen theorem [Baur-Strassen (1983)]—the only salient difference here is that this reference uses terminology from arithmetic circuits (e.g. gates, edges), instead of neural networks (e.g. nodes, weights). We will also add to Section 4 a formal definition of a standard feedforward neural network (i.e. a sequence of linear operators, followed by entrywise nonlinearities from some specified set of activation functions).
> > > > >
> > > > >
> > > > > The proposed restatement of the Lemma is as follows:
> > > > >
> > > > > ***(updated) Lemma 7***: Consider a neural network $g:\mathbb{R}^m \rightarrow \mathbb{R}$ with at most $N$ parameters and differentiable activation functions in the set $\{\sigma_i\}_{i=1}^A$. There exists a neural network of size $O(N)$ and activation functions in the set \{$ \sigma_i, \sigma'_i $\}$\_{i=1}^A$, $\rho(x) := x^2$ that calculates the gradient  $\frac{\partial g}{\partial i}$ for all $i \in [m]$.
> > > > >
> > > > >
> > > > > ***Proof***: Our proof will closely follow the proof in Theorem 9.10 in [Chen et al (2010)].
> > > > >
> > > > > By way of notation, let us denote by $g_1, g_2, \dots, g_k$ the functions calculated at each of the intermediate nodes of $g$, where $k \leq N$ is the total number of intermediate nodes, sorted from “bottom to top” (i.e. $g_k = g$). Consider the sequence of functions $h_1, h_2, \dots, h_k$ each of which calculate $g$, but $h_i$ is viewed as a function of $x_1, x_2, \dots, x_m, g_1, g_2, \dots, g_i$: namely, $h_k$ is defined as $g_k$ and $h_i$ is defined by taking $h_{i+1}$ and replacing node $i+1$ by $g_{i+1}$.
> > > > >
> > > > > We will build a new neural network with the required size and activations by adding nodes and edges (weights), as needed, to the original network, one that calculates every $\frac{\partial h_i}{\partial g_i}, i \in [k]$. We will do this in descending order of $i$. Note that $\frac{\partial h_k}{\partial g_k} = 1$, so this calculation is trivial. Then, if we have already added nodes computing $\frac{\partial h_i}{\partial g_i}, i \in [s+1, N]$, we will add nodes (and edges (weights) as needed) to compute $\frac{\partial h_s}{\partial g_s}$.
> > > > >
> > > > > By chain rule and the definitions of $h_i$, we have:
> > > > > $$\frac{\partial h_s}{\partial g_s} = \sum_{g_l \in \mbox{parents}(g_s)} \frac{\partial g_l}{\partial g_s} \frac{\partial h_l}{\partial g_l}$$
> > > > >
> > > > > Since we are proceeding in a top-to-bottom fashion, we have already added a node calculating $\frac{\partial h_l}{\partial g_l}$. We can also add a node calculating $\frac{\partial g_l}{\partial g_s}$. To do this, note we have $\frac{\partial g_l}{\partial g_s} = W_{ls} \sigma’(\tilde{g}_l)$, where $W\_{ls}$ is the weight connecting nodes $l$ and $s$ and $\tilde{g}_l$ is the pre-activation value of $g_l$. We only require a node calculating $\sigma’(\tilde{g}\_l)$ and another node to calculate the scalar product with $W\_{ls}$. Finally, we can calculate $\frac{\partial g_l}{\partial g_s} \frac{\partial h_s}{\partial g_s}$ by adding one more node with inputs the nodes $\frac{\partial g_l}{\partial g_s}, \frac{\partial h_s}{\partial g_s}$ and a square activation function. (The square activation function is so that we can write multiplication in terms of squaring, as in Lemma 8.)
> > > > >
> > > > > The total number of nodes and edges we added in this step for building this network (i.e. to calculate $\frac{\partial h_s}{\partial g_s}$) is at most a constant factor of the total number of parents $N_s$ of node $s$ in the original network. Note that $\sum_s N_s = E$, where $E$ is the total number of weights in the original network—as each weight is counted once—the new network we constructed has at most $O(N)$ weights, and uses activation functions in the set $\Sigma, \Sigma’, \rho$.
> > > > >
> > > > >
> > > > > The network also contains **all** derivatives of $g$ as internal nodes—so the claim of the lemma follows.
> > > > >
> > > > > QED
> > > > >
> > > > >
> > > > > We thank you for helping to improve the legibility of the paper and hope that our proposed modifications fully address your concerns.
> > > > >
> > > > > ---
> > > > > [1] Bai, Y., Ma, T., and Risteski, A. “Approximability of discriminators implies diversity in gans,” International Conference on Learning Representations (ICLR), 2018.
> > > > >
> > > > > [2] Agarwal, N., Allen-Zhu, Z., Bullins, B., Hazan, E., & Ma, T.  "Finding approximate local minima faster than gradient descent." Proceedings of the 49th Annual Symposium on Theory of Computing (STOC), 2017.
> > > > >
> > > > > [3] Chen, X., Neeraj K., and Wigderson, A.. "Partial Derivatives in Arithmetic Complexity and Beyond." Theoretical Computer Science 6.1-2 (2010): 1-138
> > > > >
> > > > > [4] Baur, W., & Strassen, V. (1983). The complexity of partial derivatives. Theoretical computer science, 22(3), 317-330.

---

> > > > > > ### Comment · Reviewer_rCSp · 2021-08-12
> > > > > > **Re: Re: Lemma 7**
> > > > > >
> > > > > > Thank you, this addresses my concern regarding correctness of the result about the neural network size. The choice of activation functions is quite unusual but with this large set of activation functions the result is correct.

---

> ### Author Response · Authors · 2021-08-17
> **Thank you for the discussion so far**
>
> Thanks again for engaging with us to resolve your concerns regarding the validity of Lemma 7! We promise to implement all discussed improvements in the final version. Since this issue was the primary justification for assessing the result as flawed and assigning a low score, we hope that you might consider raising your score.
>
> Additionally, now that we are on the same page regarding the Lemma, we hope you might revisit our main result and reevaluate its significance. To reiterate, our paper is the first to (i) relate the capacity required to express the solution to a linear elliptic PDE to that required to express its coefficients, while not incurring an exponential dependency on the dimension or the volume of the domain; (ii) introduce a new proof technique that simulates gradient descent in function space by growing a sequence of neural network where each iterate combines the previous iterate and the coefficients through addition, multiplication, and differentiation operators.
>
> Please let us know if there are any other questions that we can help to resolve.

---

> ### Author Response · Authors · 2021-08-20
> **In regards to choice of activations**
>
> Thank you for continuing to engage with us and for increasing your score.
>
> **Regarding the choice of activation functions**: we would like to emphasize a key point from our *general reply* to all reviewers—using standard techniques from approximation theory  [e.g. Hornik et al (1990), Yarotsky (2017]], one can approximate a neural network with one choice of nonlinearity via a (comparably sized) neural network with another choice of nonlinearity, under very mild conditions on the nonlinearities. Crucially, this simulation only increases the size by a *dimension-independent factor*. This result frees us (for purposes of deriving an expressibility result) to work with activation functions chosen for mathematical convenience and produce results that hold without loss of generality.
>
> To make this point more explicit in the paper, we propose to add the lemma below, and include the proof in this response (the lemma is written for ReLU activation functions, but proofs for other activations like sigmoid or tanh can be written completely analogously).  We note that this proof is almost verbatim the same as the proof of Lemma 1.3 in [Telgarsky (2017)].
>
> We hope this alleviates the concerns the reviewer has regarding the choice of activation functions in the main result.
>
> ***Lemma***: Let $\Omega \subseteq [-M,M]^d$ and let $G_1: [-M,M]^d \to \mathbb{R}$ be a neural network with at most $l$ layers and $n$ parameters, s.t., the weights $W^{(i)}$ for each layer $i$ and node $j$ in $G_1$ are bounded as, i.e., $\forall i,j \sum_{k} |W^{(i)}\_{jk} | \leq B $. Furthermore, assume that the activation functions used in $G_1$ belong to the set $\Xi$, s.t. all functions $\sigma: \mathbb{R} \to \mathbb{R}$ satisfy $\sup_{x \in [-B \cdot M, B \cdot M]} \sigma < M$ and have a Lipschitz constant $L$. Then there exists a neural network $G_2$ with ReLU activation and $O(n \frac{(LB)^llBM}{\epsilon} \log(\frac{(LB)^llBM}{\epsilon}))$ parameters, such that for all $\sup_{x \in [-M,M]^d} |G_1(x) - G_2(x)| \leq \epsilon$.
>
>
> ***Proof***:
> For any $\sigma \in \Xi$, from Theorem 1 in [Yarotsky (2017)] it follows that there exists a neural network $R$ with ReLU activations and $O(\frac{LBM}{\epsilon'}\log(\frac{LBM}{\epsilon'}))$ parameters such that $\sup_{x \in [-B \cdot M, B \cdot M]} \lvert \sigma(x) - R(x)\rvert \leq \epsilon'$.
>
> We will construct the network $G_2$ by replacing each activation in $G_1$ with the corresponding network $R$ as given by the result above with $\epsilon' = \epsilon/l$. Note, this network is at most a factor of $O(\frac{(LB)^llBM}{\epsilon} \log(\frac{(LB)^llBM}{\epsilon}))$ bigger than $G_1$, as the lemma requires.
>
> We will prove the claim of the lemma by induction on $l$. More precisely, we will show (by induction) that for each node at layer $i$, the network $G_2$ calculates a function that is $(LB)^i i \epsilon'$ away in $l_{\infty}$ norm from the corresponding node in $G_1$, and the inputs to the node are in $[-BM,BM]$.
>
> For the base case $i=1$, since the input $x \in [-M, M]^d$, the result follows by Theorem 1 in [Yarotsky (2017)].
>
> We proceed to the inductive claim. Let $H(x)$ denote the vector valued mapping computed by the nodes at layer $i$, and let $H_R(x)$ be the corresponding vector in $G_2$. As inductive hypothesis, we assume that $\lVert H(x) - H_R(x)\rVert_{\infty} \leq (LB)^i i\epsilon'$ for all $x \in [-M,M]^d$ and $\lVert H(x)\rVert_{\infty} \leq M$ as well as $\lVert H_R(x)\rVert_{\infty} \leq M$.
>
> Therefore, for the $j^{th}$ node in layer $(i+1)$  in network $G_1$ we have $\lvert W_j^TH(x)\rvert  \leq \lVert W_j\rVert_1\lVert H(x)\rVert_{\infty} \leq BM$ and $\sigma$ is bounded by $M$ on this interval, so we have  $\lVert \sigma_1(W_j^TH(x))\rVert_{\infty} \leq M$. Along with the bound on the activations, the part of the inductive hypothesis about the size of the input is proven.
> To prove the error bound, we have:
>
> \begin{aligned}
> \lvert \sigma_1(W_j^TH(x)) - R(W_j^TH_R(x)) \rvert  &\leq \lvert \sigma_1(W_j^TH(x)) - \sigma_1(W_j^TH_R(x)) \rvert + \lvert \sigma_1(W_j^TH_R(x)) - R(W_j^TH_R(x)) \rvert  \newline
>         &\leq L \lvert W_j^T(H(x)-H_R(x)) \rvert + \epsilon’  \newline
>         &\leq L \lVert W_j \rVert_1 \lVert H(x)-H_R(x) \rVert_{\infty} + \epsilon’  \newline
>         &\leq (LB)^{i+1}(i+1)\epsilon'
> \end{aligned}
>
>
> This finishes the proof of the inductive step, and thus the lemma.
>
> ---
> [1] Yarotsky, Dmitry. "Error bounds for approximations with deep ReLU networks." Neural Networks 94 (2017): 103-114.
>
> [2] Telgarsky, Matus. "Neural networks and rational functions." International Conference on Machine Learning. PMLR, 2017.
>
> [3] Hornik, Kurt, Maxwell Stinchcombe, and Halbert White. "Universal approximation of an unknown mapping and its derivatives using multilayer feedforward networks." Neural networks 3.5 (1990): 551-560.

---

> > ### Comment · Reviewer_rCSp · 2021-08-24
> > **Re: In regards to choice of activations**
> >
> > Thank you for this further clarification! I appreciate it!
> >
> > However, I have one more question/comment:
> >
> > The normalization assumption on the weights is quite restrictive and allows only small and/or sparse coefficients, especially in high dimension. The PDE approximating network is constructed by repeatedly applying a differential operator and various other operations to the networks that approximate the diffusion coefficient and the forcing term. The question is now if (or, under what assumptions) it is justified to assume that the final network satisfies this normalization assumption? If the weights of the PDE approximating network do not decay according to this assumption, the implicit constant in your Lemma may become quite large and potentially re-introduce a curse of dimensionality?

---

> > > ### Author Response · Authors · 2021-08-25
> > > **Re: In regards to choice of activations**
> > >
> > > Thank you for your reply. We greatly appreciate all the feedback so far, and we are glad our clarifications are helping!
> > >
> > > The weight normalization is also mostly a matter of convenience, as it makes the proof less notation-heavy (e.g. one doesn’t have to introduce an extra parameter for the bounds on the sums of the weights in the statement). We **edited** the lemma statement and proof in our previous reply to a more general version, which allows for weights summing to more than 1.
> > >
> > > Furthermore, it is possible to prove an “end-to-end” result bounding the size required to approximate the network $u_{\epsilon}$ produced by our construction *without* a worse dependence on the dimension than our Theorem 1, though there will be (naturally) dependence on other quantities like the weights in the PDE coefficient networks (more precisely, the maximum sum of weights coming in and going out of a node in the network), the maximum depth of these networks, and their Lipschitz constants. The intuitive reason why the sum of the weights appears is that all the operations in our proof (addition, multiplication, backpropagation through Lemma 7) do not create nodes with weights into and out of a node bigger than the original network.
> > >
> > > To give you a sense of the kind of result that can be obtained, we include a fully formal Corollary of Theorem 1 and the lemma in our previous reply below --- in which we bound the multiplicative factor increase in the size of the network by roughly $(LB)^{c^{T} l}$, where $L$ is a bound on the Lipschitz constant of the activations in the coefficient networks, $B$ is a bound on the sum of the weights into/out of a node in the coefficient networks, $l$ is a bound on the depth of the coefficient networks, and $c \leq 5$ is an absolute constant. (Note, there is no dependence on dimension. If we set $T = \log(1/\epsilon)$, the previous expression is roughly $(LB)^{O(1/\epsilon)}$.)
> > >
> > > We stress that we made no serious effort to optimize the dependencies on these extra parameters, as this is somewhat orthogonal to the main thrust of our paper, and it’s quite plausible that tighter bounds can be proved.
> > >
> > > ***Corollary 1***: Assume that the maximum depth of the neural networks $\tilde{A}, \tilde{c}$ and $u_0$ is $l$ and satisfy for each layer $i$ and node $j$: $\sum_k |W^{(i)}\_{j,k}| \leq B$ and $\sum_k |W^{(i)}\_{k,j}| \leq B$ for $B \geq 2$ (i.e. the “in-weights” and “out-weights” of each node are bounded by $B$). With $\tilde{\epsilon}, \epsilon, T$ defined as in Theorem 1, there exists a neural network $v\_{\epsilon}$, s.t.,
> > > - $v\_{\epsilon}$ uses ReLU activations only
> > > - $\lVert v\_{\epsilon} - u\_{\epsilon}\rVert \_{\infty} \leq \epsilon$ and
> > > - $v\_{\epsilon}$ has $O(N_T \frac{(LB)^D DBM}{\epsilon} \log(\frac{(LB)^{D} DBM}{\epsilon}))$ parameters where  $D=O(c^{T} l)$ and $N\_T = d^{2T} (N\_0 + N\_A + T(N\_f + N\_c))$. [*Note, $N\_T$ is the size bound obtained in Theorem 1*]
> > >
> > > ***Proof***
> > > First, we show the following: (i) the network $u_{\epsilon}$ satisfies $\forall i,j: \sum_k |W^{(i)}_{j,k}| \leq B$ and (ii) has depth bounded by $D=O(c^{T} l)$.
> > >
> > > To show (i), we will show that each of the operations we employ (addition, multiplication, taking derivatives) maintains this condition. Notice that multiplication and addition each *add* one node, with 2 incoming weights bounded by 1. Since $B \geq 2$, the claim obtains for these operations. Continuing to differentiation, the construction in Lemma 7 (the backpropagation lemma) constructs a network that has two copies of each of nodes in the original network: one for the “forward” network, and one for the “backward” network (in our notation, the latter nodes are $\frac{\partial h_s}{\partial g_s}$). The first types of nodes have exactly the same children as the original network, so for those nodes $v$ we have $ \sum_{k \in \mbox{child of v}} |W_{v,k}| \leq B$. On the other hand, for the latter kinds of nodes, the children of the node are the *parents* of the node in the original network. Since in our assumptions, we also assumed $\forall i,j: \sum_k |W^{(i)}\_{k,j}| \leq B$, for these nodes too we have $ \sum\_{k \in \mbox{child of v}} |W\_{v,k}| \leq B$. Thus, differentiation also maintains the bound $B$, proving (i).
> > >
> > > Now, we can apply the lemma from the previous reply to produce a network $v_{\epsilon}$ that has size $O(N_T \frac{(LB)^D DBM}{\epsilon} \log(\frac{(LB)^{D} DBM}{\epsilon}))$ where $D=O(c^{T} l)$ and $N_T = d^{2T} (N_0 + N_A + T(N_f + N_c))$.

---

> > > > ### Comment · Reviewer_rCSp · 2021-08-25
> > > > **Re: Re: In regards to choice of activations**
> > > >
> > > > I would like to again express my appreciation for your prompt reply.
> > > >
> > > > One (hopefully last) question/comment:
> > > >
> > > > The term $d^{2T}$ in the expression for $N_T$ is not bounded by a polynomial in $(d,\varepsilon^{-1})$, right? Therefore, according to some definitions, his does not strictly  break the curse of dimensionality. While I do not think that this is a huge problem, it might be good to state it.

---

> > > > > ### Author Response · Authors · 2021-08-25
> > > > > **Thank you for all the helpful feedback and raising your score**
> > > > >
> > > > > Thank you for all your helpful feedback, as well as raising your score. It will make the camera ready version of the paper substantially stronger and clearer!
> > > > >
> > > > > Regarding the last question: to approximate $u^*$ within $\epsilon$, the required $T$ is roughly $O(\log(1/\epsilon))$ (treating $\kappa, R$ as constants)—hence the scaling with dimension is roughly $d^{O(\log(1/\epsilon))}$. If one thinks of the desired accuracy $\epsilon$ as a constant (which seems sensible in many settings), this is polynomial in $d$. In order to get an exponential dependence in $d$ (i.e. what we would traditionally think of as “curse of dimensionality”), $\epsilon$ would have to be exponentially small in $d$.
> > > > >
> > > > > We take your point, however, and are more than happy to make this point more explicit in the exposition in the paper. Furthermore, we propose to also modify the statement of the informal theorem in line (64) as:
> > > > >
> > > > > **Theorem (Informal)**. If the coefficients $A$, $c$ and the function $f$ are approximable by neural networks with at most $N$ parameters, the solution $u^*$  to the PDE in Definition 1 is $\epsilon$-approximable by a neural network with $O (d^{O(\log(1/\epsilon))} N)$ parameters.

---

### Official Review · Reviewer_gsRU · 2021-07-16

**Rating:** 8
**Confidence:** 5

**Summary:**

The paper proved an $\epsilon$ approximation theory showing that the neural network approximation for linear elliptical PDEs is bounded by the number of NN parameters required for approximating the PDE coefficients as well as the dimension of the domain. The proof is mainly based on an operator theoretic point of view and the bounds are obtained by approximating gradient descent using neural networks. The results also provide insights into the recent success of using NN to approximate high dimensional as well as parametric PDEs.

**Limitations And Societal Impact:**

yes

**Main Review:**

The paper provides a rigorous proof for an $\epsilon$ upper-bound for the approximation error. The paper provides important insights on the success of the NN approximation of high-dimensional/parametric PDEs. I enjoyed reading the paper and believe that it will be of interest to the NeurIPS audience. I therefore recommend a clear accept.

Below are some issues/questions that I have:
(1)   Section 4 provides a list of well-known definition/theorem in functional analysis. While I appreciate the author's effort of including these in the main section, they should at least provide the references for the definitions/theories as they are not the authors' work.
(2)   The proof is mainly based on the fact that the gradient descent algorithm can be approximated by an NN. I wondering in practice, will this hold true? In other words, how tight are these bounds? Given the fact that the paper mainly concerns linear elliptic PDEs one should be able to easily check the scaling proposed in the main theorem.
 (3)  The paper mainly concerns the PDE coefficient $a(x)$ and the source term $f(x)$. How would the main conclusion change when the boundary condition is non-trivial?

**Time Spent Reviewing:**

4

---

> ### Author Response · Authors · 2021-08-10
> **Reply to Reviewer gsRU**
>
> We thank the reviewer for their positive review and feedback! Regarding missing references for the functional analysis definitions, we will make sure that they are added to the main paper in the camera ready version.
>
> **Re: if the gradient descent approximation holds in practice and if the bounds are tight**\
> This is a very interesting question! Verifying the empirical performance of the architecture in our construction (in particular, the implied weight-tying) seems like fertile ground for future work. As mentioned in Section 8, we believe that the bounds can be further improved and since we do not have a matching lower bound, we can’t comment on how tight our bound is (this is also an intriguing open question). We note that empirically checking whether the scaling of our bounds is tight seems challenging: one would need some way to (near-)exhaustively check all PDEs in a certain family.
>
> **Re: How would the main conclusion change when the boundary condition is non-trivial**\
> That is an interesting question. We focused on the paper on the homogeneous Dirichlet boundary condition, i.e., $u(x) = 0$ for all $x \in \partial \Omega$. However, it seems to us that extending our work to the non-homogeneous Dirichet boundary condition, i.e., when $u(x) = g(x)$ for all $x \in \partial \Omega$ is possible, with appropriate technical assumptions on $g$.
> Namely, there is a standard reduction to the homogeneous case, which we recap here. Let us assume there exists a function  $G \in H^1(\Omega)$, s.t. $G(x) = 0$ for all $x \in \Omega$ and $G(x) = g(x)$ for all $x \in \partial \Omega$. By denoting $U = u - G$, we readily see that $LU = f - LG$ for all $x \in \Omega$ and $U = 0$ for all $x \in \partial \Omega$—so it suffices to solve this homogeneous problem. If $G$ and $LG$ are sufficiently smooth and approximable by neural networks, it seems like the techniques in our paper should carry over.

---

> > ### Comment · Reviewer_gsRU · 2021-08-25
> > **Regarding the non-trivial boundary condition**
> >
> > The author have addressed all my concerns. While I agree with the author's statement with the Dirichlet boundary condition, it might not be a trivial task to extend this to other type of boundary conditions (Neuman/Robin). However I think this is out of the scope of the current paper and I appreciate the author's effort on clarification.

---

### Official Review · Reviewer_PQMY · 2021-07-21

**Rating:** 6
**Confidence:** 3

**Summary:**

The authors show that the solution of a linear elliptic PDE (with boundary condition) can be represented with a neural network, whose number of parameters is at most polynomial in the input dimension d, under the assumption that the coefficients of the PDE are given by NNs with poly(d) parameters.

This is achieved by first showing that gradient descent with a particular choice of stepsize, provides a sequence in $L_2$, that converges to the solution of the PDE. Then the iterates of the sequence are approximated by NNs in a recursive manner, and counting the number of parameters that are added at each iteration.

In essence, the result aims to help the understanding of how learning a solution of a high-dimensional PDE from samples might be computationally feasible using parametrized families of functions, like NNs.

**Limitations And Societal Impact:**

 addressing potential negative societal impact is not relevant to this work. The authors do not discuss the limitations of their work.

**Main Review:**

The **originality** of this work hinges on its differences with the reference [14], which proves a similar result. Unfortunately I am not familiar with that paper. In this work, the authors only present a brief comparison to [14] claiming that "their rate depends on the volume of the domain, and thus can have an exponential dependence on the dimension", which would set them apart as they find mild conditions for the soution to require poly(d) parameters. After a quick glance at [14], indeed their results depend on the volume of the domain, but I believe it would be a good Idea to discuss reference [14] in depth, in the prior work section. In any case, reference [14] appears to provide a less general result that focuses on the Poisson equation, and the method of proof is vastly different.

The main issue I have found is regarding the **clarity**. The main result **Theorem 1** has an unspecified parameter $\gamma$ that does not even appear in the paper up to that point. This makes it extremely hard to have a clear grasp of the implications of the theorem, because if $\gamma \to \delta^+$ then $\tilde{\epsilon} \to \infty$ and the bound presented becomes vacuous. After some effort, it appears that the definition of $\gamma$ appears for the first time in **Lemma 2**, 2 pages after **Theorem 1**! All parameters involved in **Theorem 1** should have a clear definition at the moment it is stated.

Moreover, if $\gamma$ is in fact like in **Lemma 2**, It is a quantity related to the operator $L$ and the results only make sense, I believe, if $\gamma > \delta$. So in fact there is a hidden assumption that is not disclosed nor discussed. Can it happen that $\delta = \gamma$, for example? why not? does it matter? This should be discussed and explained by the authors.

Another point that is not **clear** is what particular property of neural networks are used in the proofs. I have to disclose that I did not delve into the details of the supplementary material, but in the main text I think there are no particular properties of neural networks that are used to obtain the results. Everything is stated in number of parameters needed to approximate a function. Do the results hold for other classes of parametric functions like say, linear over features, products of polynomials, etc? It seems like the arguments using the gradient descent sequence and backpropagation should also work. If there are no particular properties of neural networks that are used in the proofs then It seems like it is just a term that is being thrown in without need. After briefly checking the supplementary It seems that only Lemma 8,9 talk about NNs but they just seem like general properties of automatic differentiation (that should apply to other classes of parametric functions) (Lemma 8) or allowing to take the square of the function (Lemma 9) which leads to the mostly artificial assumption that the quadratic activation function should be allowed in Theorem 1 (Line 186).

Finally regarding **clarity**, the term *small neural network* is used throughout but the meaninig of *small* is not actually clear until later in the paper. It would be better if this can be clarified as early as possible otherwise the meaning of some claims in the abstract/introduction are too vague.

The synthetic assumption of a quadratic activation function being allowed in the neural network in Theorem 1 has practical implications that are not clear. Does it mean that when solving a PDE using a NN one should put the quadratic activation in some neurons? in all? try all combinations? The theorem, of course, only talks about existence of a solution and practical implications seem to be out of the scope of this work. Nevertheless this is an interesting point to discuss.

The **quality** of the work would be better if the previously mentioned issues are clarified/improved. Other than that, the paper is overall well written and the authors show expertise in the topics discussed.

The theoretical results seem **significant** as the apparent absence of the curse of dimensionality for NNs solutions to PDEs is not understood. However, I find it hard to see practical impacts of the work, other than my observation that using some quadratic activations might improve performance.

**Time Spent Reviewing:**

7

---

> ### Author Response · Authors · 2021-08-10
> **Reply to Reviewer PQMY**
>
> We thank the reviewer for the reviewer and the positive assessment and are glad that you think that our result is significant!  Please find our replies to your comments below:
>
> **Re: in-depth discussion of [14] in prior work**\
> We thank the reviewer for their suggestion—we will add a deeper discussion of [14] and comparison with our methods to the prior work section.
>
> **Re: defining $\gamma$ before Theorem 1** \
> Thank you for the suggestion! We will move the definition of γ before the statement of Theorem 1 in the camera ready version of the paper.
>
> **Re: Clarification regarding value of $\gamma$** \
> The results indeed implicitly assume $\gamma > \delta$. This regime of parameters can be justified as follows: $\gamma$ is a property of L (i.e. the particular PDE being solved), whereas we think of $\delta$ as small (and tending to zero), given that we are interested in PDEs in which the neural networks $\tilde{A}$ and $\tilde{c}$ approximate the coefficients $A$ and $c$ well. We will add an explicit note in the paper that $\gamma > \delta$.
>
> **Re: no particular property of neural networks being used in the proof** \
> Perhaps we could have made this clearer in the paper: in fact the proofs are fundamentally tied to the properties of neural networks. Crucially, we leverage the following properties (see line 277):
> 1. Adding two neural networks results in a neural network whose size equals the sum of the two input networks.
> 2. Multiplication of two neural networks results in a neural network of size equal to the sum of the two networks (by adding a square activation).
> 3. The derivative of a neural network is a neural network of size at most a constant factor bigger (using Lemma 7).
>
> **Re: “... the meaning of __small__ is not clear until later in the paper …”**\
> Thanks for raising this concern. We will improve the exposition to make our meaning clearer at the outset. We considered the alternative terminology “bounded” but in the end felt that “small” better reflects the thrust of the main results: the interesting regime for our results is when the neural networks have subexponential size. (Otherwise more classical universal approximation results give comparable bounds.)  We will add the appropriate explanation at the first mention in the introduction.
> We also note that this ambiguity only exists in the prose. Our theorem statements and proofs are all articulated formally.
>
> **Re: practical implications of quadratic activations**\
> Please see our elaboration on this point in the general reply to all reviewers. In short, the choice of activations is immaterial to the final results and is for mathematical convenience. We also discuss the practical takeaways from the paper at length in the general reply.

---

> > ### Comment · Reviewer_PQMY · 2021-08-20
> > **About the properties of neural networks used in the proof.**
> >
> > thank you for your answer. I think my question was not well understood. I would like to know what properties of neural networks, **not shared with other classes of functions**, are used in the proof. that is, what is unique about neural networks here? for example take the class of polynomials, then
> >
> > 1. adding two polynomials results in a polynomial of size equal to the sum of the parts
> > 2. multiplication, the same.
> > 3. I think lemma 7 comes from the more general result about reverse mode automatic differentiation, so if a polynomial can be evaluated in time T, then its derivative can also be evaluated in time O(T), so the derivative also has some similar size.
> >
> > this is just a quick example to illustrate my question.

---

> > > ### Author Response · Authors · 2021-08-21
> > > **Re: About the properties of neural networks used in the proof**
> > >
> > > Thank you for your clarification. It is plausible that our techniques are compatible with other function classes that satisfy a similar “complexity control” under operations of addition, multiplications and differentiation. We think this is a feature of our proof techniques, rather than a bug! It seems difficult however to state a general “meta-theorem” for this proof framework, of which the neural network result is a special case—which is why the results don’t mention any other function classes beyond neural networks.
> > >
> > > We note that it’s fairly natural for a proof technique to apply to more than just one class of functions—indeed, a theorem that *uniquely* applied only to neural networks would be unusual, and we might be skeptical that actually held—in fact, we do not know of such results in the approximation theory literature. (Such a result would also need to strongly depend on the architecture of the network class—since, e.g., *a neural network with polynomial activations is a polynomial*.)
> > >
> > > Regarding your proposed example: it is unclear what the “size” of a polynomial is—we assume the reviewer has in mind the number of nonzero coefficients (another standard measure of complexity in the approximation theory literature is the degree of the coefficient). With this interpretation, multiplying two polynomials with $N$ and $M$ nonzero coefficients results in a polynomial with $NM$ nonzero coefficients; taking a derivative of a polynomial over d variables with $N$ nonzero coefficients results in a polynomial with up to $Nd$ nonzero coefficients. Thus, a back-of-the-envelope calculation suggests that one could plausibly use similar techniques to ours to prove a complexity bound (i.e. a bound on the number of nonzero coefficients) of roughly ~ $\max(C,d)^{\log(1/\epsilon)}$ where $C$ is a bound on the number of nonzero coefficients in the polynomials constituting the PDE coefficients $a, f$.

---

> ### Author Response · Authors · 2021-08-30
> **Can we help address some other concerns?**
>
> We thank you for the discussion so far. Did our reply help address your concerns? Please let us know if there are some other questions that we can resolve.

---

### Author Response · Authors · 2021-08-10
**General Reply to All Reviewers**

We thank the reviewers for their feedback and the detailed comments.

We are encouraged to see that two reviewers advocate for acceptance, noting the significance our theoretical results (Reviewer gsRU), the importance of our insights regarding the success of the neural networks in approximating solutions of high-dimensional PDEs (Reviewer PQMY) and that the paper is well written.

While Reviewers rCSp and WhbH assigned low scores, we believe that both assessments owe primarily to misunderstandings that can be resolved in the rebuttal period and addressed through improvements to the exposition. To summarize:

* Reviewer rCSp was concerned about a potential bug in our proof—namely that the neural networks in considerations might not be differentiable. We state however clearly as one of our main assumptions that PDEs we consider have coefficients approximable by neural networks with infinitely differentiable activations (lines 159-160).  We expand on this below and in the individual reply as well.
* Reviewer WhbH was concerned with the absence of comparisons to classical methods in cases where the PDE’s solution is highly regular (e.g., infinitely differentiable)—as we argue in more detail below, regularity alone cannot guarantee sub-exponential growth of the neural network required to approximate a function. For example, approximating an infinitely differentiable function over a compact domain may require an exponentially sized network.

Before addressing each reviewer’s individual concerns in their respective threads, we briefly address some common concerns below:

* ***Choice of activation functions in our assumptions and main theorem***: The approximating neural networks in Assumption 3 ($\tilde{A}, \tilde{c}$ and $f_{nn}$) and the network $u_\epsilon$ produced by our construction (see Theorem 1) both use infinitely differentiable activations. In particular, the set of activations used by $u_\epsilon$ consists of the derivatives of the activation functions in Assumption 3 together with the square activation function ($x^2$).

  **These assumptions are solely for mathematical convenience**: standard results in approximation theory [Yarotsky 2017, Barron 1993] show that one can approximate (in the $l_{\infty}$ sense) a neural network with one choice of nonlinearity via a (comparably sized) neural network with another choice of nonlinearity—under mild assumptions on the nonlinearity. When approximating the function over a compact domain, achieving an $\epsilon$ approximation can be done by incurring a $poly(1/\epsilon)$ or $poly(\log(1/\epsilon))$ factor increase in size (note there is *no dependence* on dimension). Thus, if the coefficients are approximable by a neural network with a non-differentiable activation function, they can also be approximated by a slightly larger network with a differentiable activation over any compact domain. Similarly, the activations on the network resulting from Theorem 1 can be replaced by a different activation at the expense of a slight blowup in size. We will make sure to add a remark to clarify this in the main paper.

* ***Practical takeaways from our work***: The key result of our paper is a bound on the size of the neural networks required to represent the solutions of PDEs whose coefficients can be approximated by neural networks. This has both computational and statistical consequences for how neural networks are used to solve PDEs: (i) **computational**: PDE solvers based on neural networks are only efficient (in particular outperform classical mesh-based methods) when the neural networks being fitted are not too big; (ii) **statistical**: many PDE solvers (e.g. deep Ritz/Galerkin based methods [E et al, 2018, Lu et al, 2021]) are based on Monte Carlo approximations of certain integrals—the effective sample complexity of which depends on the size of the network being fit. Additionally, as we note in Sections 7 and 8, our proof suggests a particular “weight-tied” architecture which may be more sample efficient in practice—we leave such empirical studies as a fertile ground for future work.

We further address each reviewer’s individual concerns in greater depth in the respective threads.

---

[1] Yarotsky, Dmitry. "Error bounds for approximations with deep ReLU networks." Neural Networks 94 (2017): 103-114.

[2] Barron, Andrew R. "Universal approximation bounds for superpositions of a sigmoidal function." IEEE Transactions on Information theory 39.3 (1993): 930-945.

[3] Weinan, E., and Bing Yu. "The deep Ritz method: a deep learning-based numerical algorithm for solving variational problems." Communications in Mathematics and Statistics 6.1 (2018): 1-12.

[4] Lu, Jianfeng, Yulong Lu, and Min Wang. "A priori generalization analysis of the deep ritz method for solving high dimensional elliptic equations." arXiv preprint arXiv:2101.01708 (2021).

---

### Decision · Program_Chairs · 2021-09-27

**Decision:**

Accept (Spotlight)

**Comment:**

Recently, neural networks (NN) have had great success in approximating solutions of partial differential equations (PDEs). One key observation is that the NN approach to PDEs do not seem to suffer from the curse-of-dimensionality. Hence, it is important to understand the strengths and limitations of this approach.

The authors take on a particular class of PDEs (i.e., linear elliptic PDEs with Dirichlet boundary conditions) and provide a theoretical characterization of how many parameters are needed to approximate their solutions within a desired accuracy. They identify "small" networks suffice where the number of parameters depend polynomially on the dimension and linearly on the number of parameters required to express the PDE. The theoretical analysis is non-trivial and the paper provides a great path moving forward in the PDE research vein.

While the initial scores of the work were below the threshold, the rebuttal was effective in clarifying the concerns of the reviewers, in particular mooting a counter-example proposed by a reviewer, confusions on the applications of the gradient operators (due to the final activations, and the way in which the NNs are constructed by authors (i.e., growing network in each iteration). As a result, the scores improved uniformly and I thank both the authors and the reviewers for their efforts.